# Recurring patterns in bacterioplankton dynamics during coastal spring algae blooms

Hanno Teeling[1][*][†], Bernhard M Fuchs[1][*][†], Christin M Bennke[1][‡], Karen Krüger[1], Meghan Chafee[1], Lennart Kappelmann[1], Greta Reintjes[1], Jost Waldmann[1], Christian Quast[1], Frank Oliver Glöckner[1], Judith Lucas[2], Antje Wichels[2], Gunnar Gerdts[2], Karen H Wiltshire[3], Rudolf I Amann[1][*]

[1]Max Planck Institute for Marine Microbiology, Bremen, Germany; [2]Biologische Anstalt Helgoland, Alfred Wegener Institute for Polar and Marine Research, Helgoland, Germany; [3]Alfred Wegener Institute for Polar and Marine Research, List auf Sylt, Germany

*For correspondence: hteeling@mpi-bremen.de (HT); bfuchs@mpi-bremen.de (BMF); ramann@mpi-bremen.de (RIA)

[†]These authors contributed equally to this work

Present address: [‡]Section Biology, Leibniz Institute for Baltic Sea Research, Warnemünde, Germany

Competing interests: The authors declare that no competing interests exist.

**Abstract** A process of global importance in carbon cycling is the remineralization of algae biomass by heterotrophic bacteria, most notably during massive marine algae blooms. Such blooms can trigger secondary blooms of planktonic bacteria that consist of swift successions of distinct bacterial clades, most prominently members of the *Flavobacteriia*, *Gammaproteobacteria* and the alphaproteobacterial *Roseobacter* clade. We investigated such successions during spring phytoplankton blooms in the southern North Sea (German Bight) for four consecutive years. Dense sampling and high-resolution taxonomic analyses allowed the detection of recurring patterns down to the genus level. Metagenome analyses also revealed recurrent patterns at the functional level, in particular with respect to algal polysaccharide degradation genes. We, therefore, hypothesize that even though there is substantial inter-annual variation between spring phytoplankton blooms, the accompanying succession of bacterial clades is largely governed by deterministic principles such as substrate-induced forcing.

## Introduction

Pelagic zones of the world's oceans seemingly constitute rather homogenous habitats, however, they feature enough spatial and temporal variation to support a large number of species with distinct niches. This phenomenon has been termed 'paradox of the plankton' by G. Evelyn Hutchinson (*Hutchinson, 1961*). Interactions within planktonic microbial communities are manifold and complex (see *Amin et al., (2012)* and *Worden et al. (2015)* for reviews). Still, planktonic microbial communities are simple in comparison to benthic or terrestrial soil communities and thus particularly suitable for the study of microbial community composition dynamics. In recent years, continuous biodiversity studies at long-term sampling stations have started to reveal discernible deterministic patterns within marine microbial plankton communities (see *Fuhrman et al. (2015)* for a recent review). This is particularly true for less dynamic oligotrophic oceanic regions that are dominated by the members of the alphaproteobacterial *Pelagibacteriaceae* (SAR11 clade) and the cyanobacterial *Prochlorococcaceae (Prochlorococcus marinus)*. By contrast, more dynamic eutrophic coastal regions are subject to frequent system perturbations and thus seldom in a state of equilibrium. This can lead to apparently stochastic changes in bacterioplankton community composition. To capture recurrence of biodiversity patterns in such coastal areas, sampling must occur at the order of weekly to sub-weekly time scales over multiple years. Owing to the lack of such intensively sampled long-term time series

**eLife digest** Small algae in the world's oceans remove about as much carbon dioxide from the atmosphere as land plants. These algae do not grow continuously, but often surge in numbers during temporary blooms. Such blooms can be large enough to be seen from space by satellites. The lifespan of algae within such blooms is short, and when they die, marine bacteria feed on the remnants, which releases much of the stored carbon dioxide.

Much of an algal cell consists of different types of polysaccharides. These large molecules are essentially made from sugars linked together. Polysaccharides are varied molecules and can contain many different sugars that can be linked in a number of different ways. During algae blooms bacteria proliferate that are specialized in the degradation of these polysaccharides. In 2012, researchers reported how over the progression of an algae bloom different groups of marine bacteria bloomed in rapid succession. However, it remained unknown whether the same or different groups of bacteria respond to algae blooms at the same place from year to year, and whether or not these bacteria use the same enzymes to degrade the polysaccharides.

Teeling, Fuchs et al. – who include many of the researchers from the 2012 study – now report on the analysis of a series of algae blooms that occurred in the southern North Sea between 2009 and 2012. The analysis is based on samples collected every week during the spring seasons, and shows that certain groups of related bacteria, known as clades, became common during each bloom. Teeling, Fuchs et al. also found indications that the clades that repeatedly occurred had similar sets of genes for degrading algal polysaccharides, but that the sets were different between the clades.

These data suggest that there is a specialized bacterial community that together can degrade the complex mixture of algal polysaccharides during blooms. This community reappears each year with an unexpectedly low level of variation. Since different species of algae made up the blooms in each year, this finding suggests that the major polysaccharides in these algae are similar or even identical.

Future work will focus on the specific activities of bacterial enzymes that are needed to degrade polysaccharides during algae blooms. Study of these enzymes in the laboratory will help to resolve, which polysaccharides are attacked in which manner, and to ultimately help to identify the most abundant algal polysaccharides. This will improve our current understanding of the carbon cycle in the world's oceans.

data, our current understanding of the extent and predictability of recurring microbial biodiversity patterns for such marine habitats is still limited.

A particularly important connection in the marine carbon cycle exists between marine microalgae as primary producers and heterotrophic bacteria that feed on algal biomass. Global photosynthetic carbon fixation is estimated to exceed 100 Gigatons yearly, of which marine algae contribute about half (*Falkowski et al., 1998*; *Field et al., 1998*; *Sarmento and Gasol, 2012*). Planktonic uni- to pluri-cellular algae such as diatoms, haptophytes, and autotrophic dinoflagellates are the most important marine primary producers. Diatoms alone are estimated to contribute 20–40% to global carbon fixation (*Nelson et al., 1995*; *Mann, 1999*; *Armbrust, 2009*).

Primary production by planktonic microalgae differs from primary production by sessile macroalgae or land plants as it is much less constant, but culminates in blooms that are often massive, as occurs worldwide during spring blooms from temperate to polar regions. These blooms are highly dynamic phenomena that are time-limited by nutrients, predator grazing and viral infections. Bloom termination results in a short-lived massive release of algal organic matter that is consumed by dedicated clades of heterotrophic bacterioplankton. This trophic connection leads to synchronized blooms of planktonic bacteria during phytoplankton blooms, as has been described in various studies (*Bell and Kuparinen, 1984*; *Niu et al., 2011*; *Tada et al., 2011*; *Teeling et al., 2012*; *Yang et al., 2015*; *Tan et al., 2015*).

The activities of these heterotrophic bacteria impact the proportion of algal biomass that is directly mineralized and released back into the atmosphere mostly as carbon dioxide, and the algae-derived biomass that sinks out to the bottom of the sea as carbonaceous particles. These are further remineralized by particle-associated bacteria while sinking and by benthic bacteria when reaching

the sediment, even in the deep sea (e.g. *Ruff et al., 2014*). The remainder is buried for a long time as kerogen and forms the basis for future oil and gas reservoirs. The ratio between bacterial mineralization and burial of algae-derived organic matter thus has a profound influence on the atmospheric carbon dioxide concentration (*Falkowski et al., 1998*). However, the bulk of bacteria during phytoplankton blooms are free-living and not attached to particles or algae. These bacteria play a pivotal role in the mineralization of algae-derived non-particulate dissolved organic matter (DOM).

The bacterial clades that respond most to phytoplankton blooms belong to the classes *Flavobacteriia* (phylum *Bacteroidetes*) and *Gammaproteobacteria*, and the *Roseobacter* clade within class *Alphaproteobacteria* (*Buchan et al., 2014*). This response is typically not uniform, but consists of a series of distinct clades that bloom one after another. In the year 2009, we investigated the response of bacterioplankton to a diatom-dominated spring phytoplankton bloom in the German Bight (*Teeling et al., 2012*). Within the free-living bacteria (0.2 to 3 µm) we observed a swift succession of bacterial clades that were dominated by *Flavobacteriia* and *Gammaproteobacteria*, with consecutively blooming *Ulvibacter (Flavobacteriia)*, *Formosa (Flavobacteriia)*, *Reinekea (Gammaproteobacteria)*, *Polaribacter (Flavobacteriia)* genera and SAR92 (*Gammaproteobacteria*) as prominent clades.

Using time-series metagenome and metaproteome analyses, we demonstrated that the substrate-spectra of some of these clades were notably distinct. The succession of bacterioplankton clades hence constituted a succession of distinct gene function repertoires, which suggests that changes in substrate availability over the course of the bloom were among the forces that shaped the bacterioplankton community. Dominance of bottom-up over top-down control is assumed to be characteristic for the initial phases of spring phytoplankton blooms. After winter, inorganic nutrients are aplenty, and the overall abundance of microbes is low. When suitable temperature and sunlight conditions are met in spring, algae and subsequently bacteria can enter an almost unrestricted proliferation. In contrast, predators such as flagellates, protists and zooplankton can only start proliferating when their food sources are available in larger numbers. Hence, top-down control by predation sets in only during later bloom phases. This situation is distinct from summer and fall phytoplankton blooms.

Pronounced differences between blooming clades were found in the gene frequencies and protein expression profiles of transporters and carbohydrate-active enzymes (CAZymes; [*Cantarel et al., 2009*; *Lombard et al., 2014*]), such as glycoside hydrolase (GH), polysaccharide lyase (PL), carbohydrate esterase (CE), or carbohydrate-binding module (CBM) containing genes. The latter indicates a pronounced niche partitioning with respect to algal polysaccharide degradation. Marine algae produce large quantities of distinct polysaccharides, for example storage, cell matrix and cell wall constituents, or as part of extracellular transparent exopolymer particles (TEP). It has been recently shown that in particular *Flavobacteriales* and *Rhodobacterales* respond to TEP availability (*Taylor et al., 2014*). The diversity of algal polysaccharides is too high for a single bacterial species to harbor all the genes required for the complete degradation of all naturally occurring variants. Thus, polysaccharide-degrading bacteria specialize on dedicated subsets of polysaccharides, which is why the decomposition of algal polysaccharides during and after algal blooms is a concerted effort among distinct bacterial clades with distinct glycan niches (e.g. *Xing et al., 2015*).

In this study, we provide evidence that the succession of bacterioplankton clades that we reported for the 2009 North Sea spring phytoplankton bloom re-occurred during the spring blooms from 2010 to 2012. We tested whether the bacterioplankton clades and their associated CAZyme repertoires differ from year to year or exhibit recurrent patterns. We analyzed spring bacterioplankton community composition via 16S rRNA catalyzed reporter deposition fluorescence in situ hybridization (CARD-FISH) and 16S rRNA gene tag sequencing, as well as gene function repertoires by deep metagenome sequencing. Our efforts have culminated into the as of yet highest resolved dataset capturing the response of planktonic bacteria to marine spring phytoplankton blooms and have allowed identification of recurring patterns that might ultimately lead to an explanatory model for bacterioplankton succession dynamics during spring algae blooms.

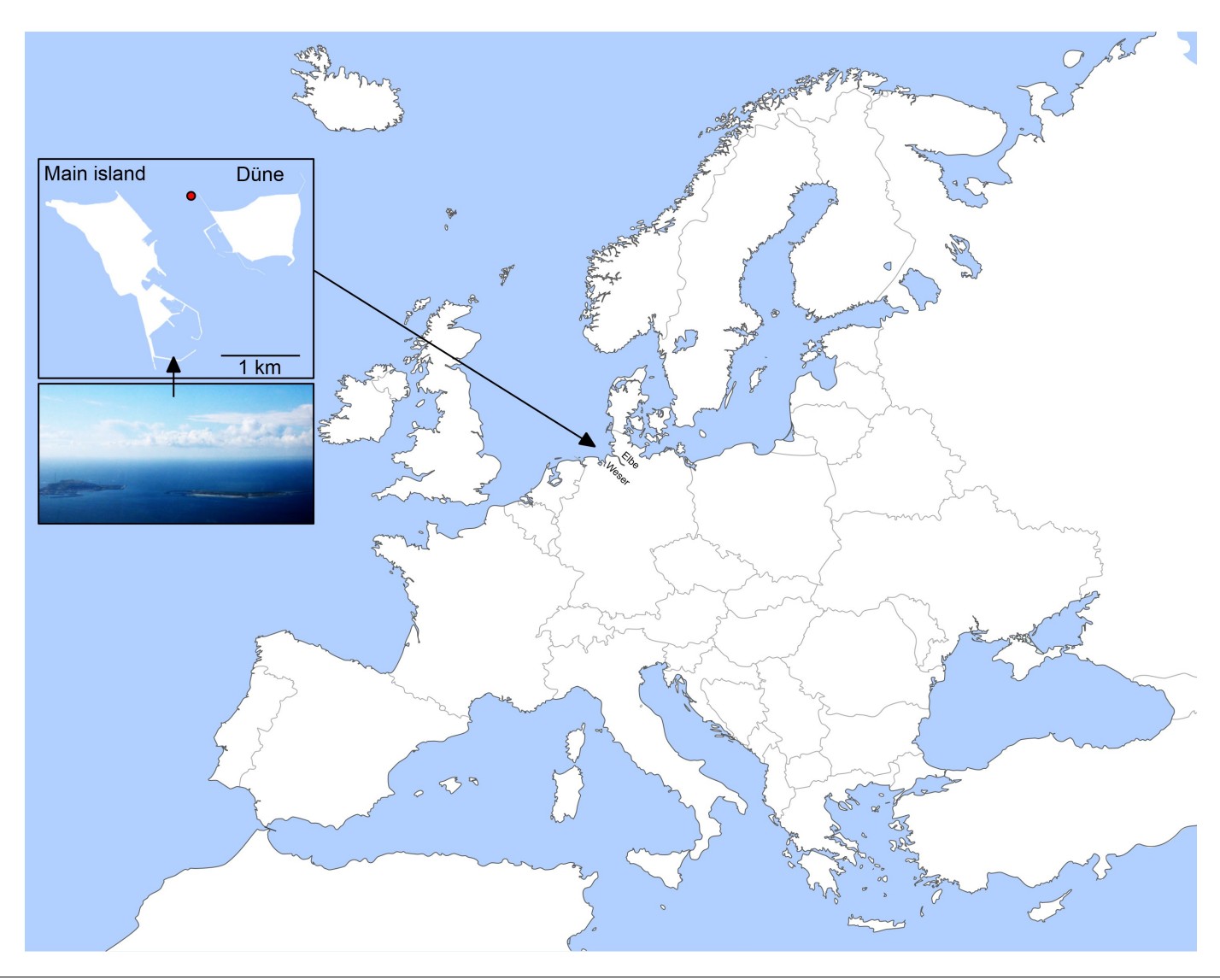

**Figure 1.** Location of Helgoland Island (ca. 40 km offshore the northern German coastline) and the long-term ecological research site 'Kabeltonne' (red circle: 54° 11.3' N, 7° 54.0' E) in the German Bight of the North Sea.

## Results

### Sampling site characteristics

The samples were taken at Helgoland Island about 40 km offshore in the southeastern North Sea in the German Bight at the station 'Kabeltonne' (54° 11.3' N, 7° 54.0' E; *Figure 1*) between the main island and the minor island, Düne (German for 'dune'). Water depths at this site fluctuate from 6 to 10 m over the tidal cycle. During most of the year, a westerly current transports water from the English Channel alongside the Dutch and Frisian coast to Helgoland, but water around the island is also influenced by nutrient inputs from the rivers Weser and Elbe and from the northern North Sea (*Wiltshire et al., 2010*). During the 2009 to 2012 study period, the lowest water temperatures were measured in mid to late February (min. 2010: 1.1°C; max. 2009: 3.4°C), followed by a continuous increase until a peak in August (min. 2011: 18.0°C; max. 2009: 18.7°C) (*Supplementary file 1*).

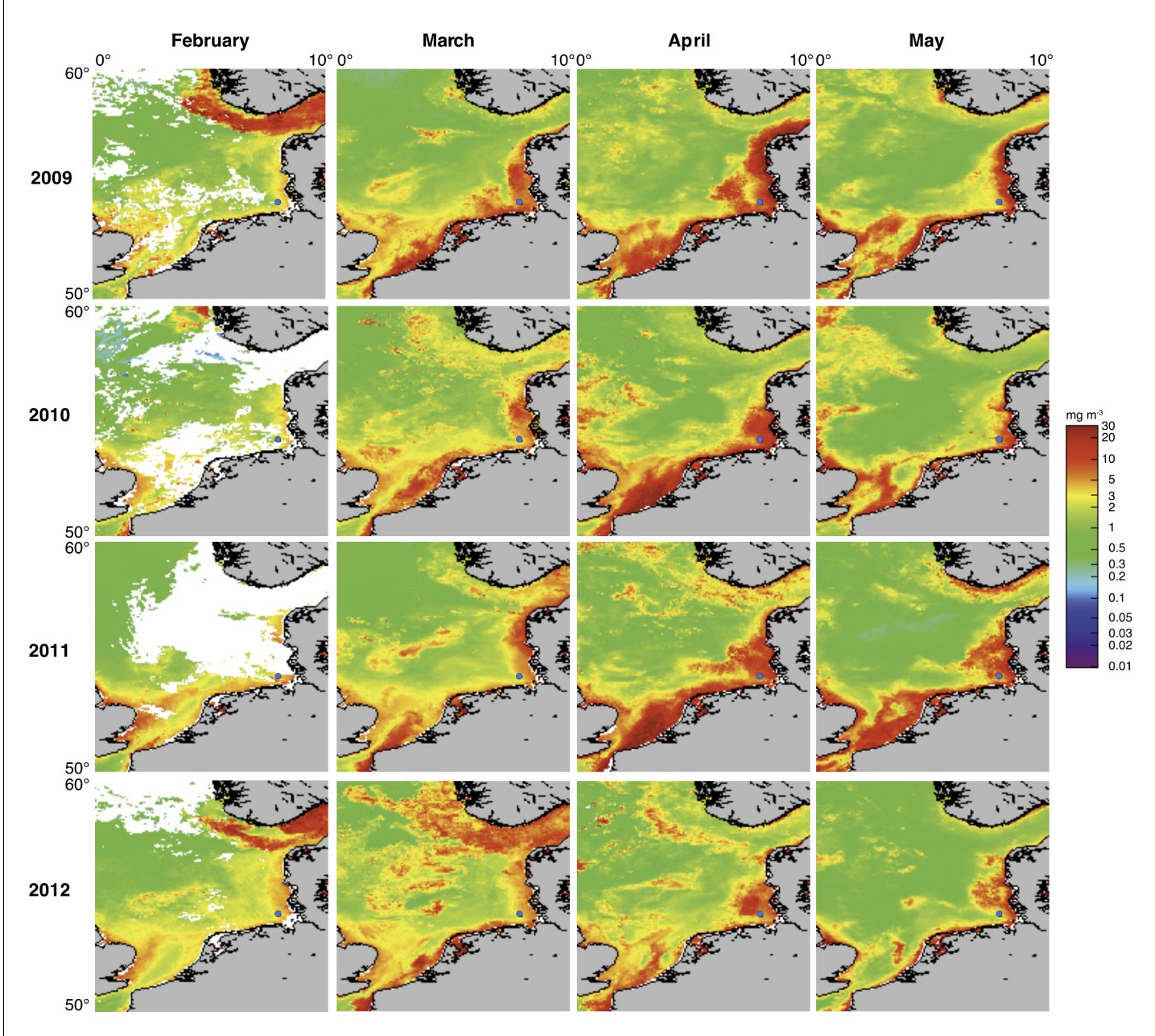

**Figure 2.** Satellite chlorophyll *a* measurements. Data are shown for the southern North Sea for the months February to May (monthly averages) of the years 2009 to 2012. Images were retrieved from the GlobColour website using the 'extended Europe area' at full resolution (1 km) as merged products of weighted averages from the following sensors: MERIS, MODIS AQUA, SeaWIFS and VIIRS. See GlobColour website for details (http://hermes.acri.fr). The position of Helgoland Island is indicated by a blue dot.

## Phytoplankton - diversity and bloom characteristics

Spring phytoplankton blooms in the North Sea typically develop during March and reach highest intensities during April and May. The highest chlorophyll *a* concentrations are usually observed at the coastlines including the area around Helgoland Island (*Figure 2*). North Sea spring blooms are thus large-scale phenomena that are, however, influenced by local conditions, such as riverine inputs. At Helgoland island, spring phytoplankton blooms started around mid March when water temperatures surpassed 3 to 5°C (*Figure 3A–H*; *Supplementary file 1*). The diatoms *Chaetoceros debilis and Chaetoceros minimus, Mediopyxis helysia, Rhizosolenia styliformis* and *Thalassiosira*

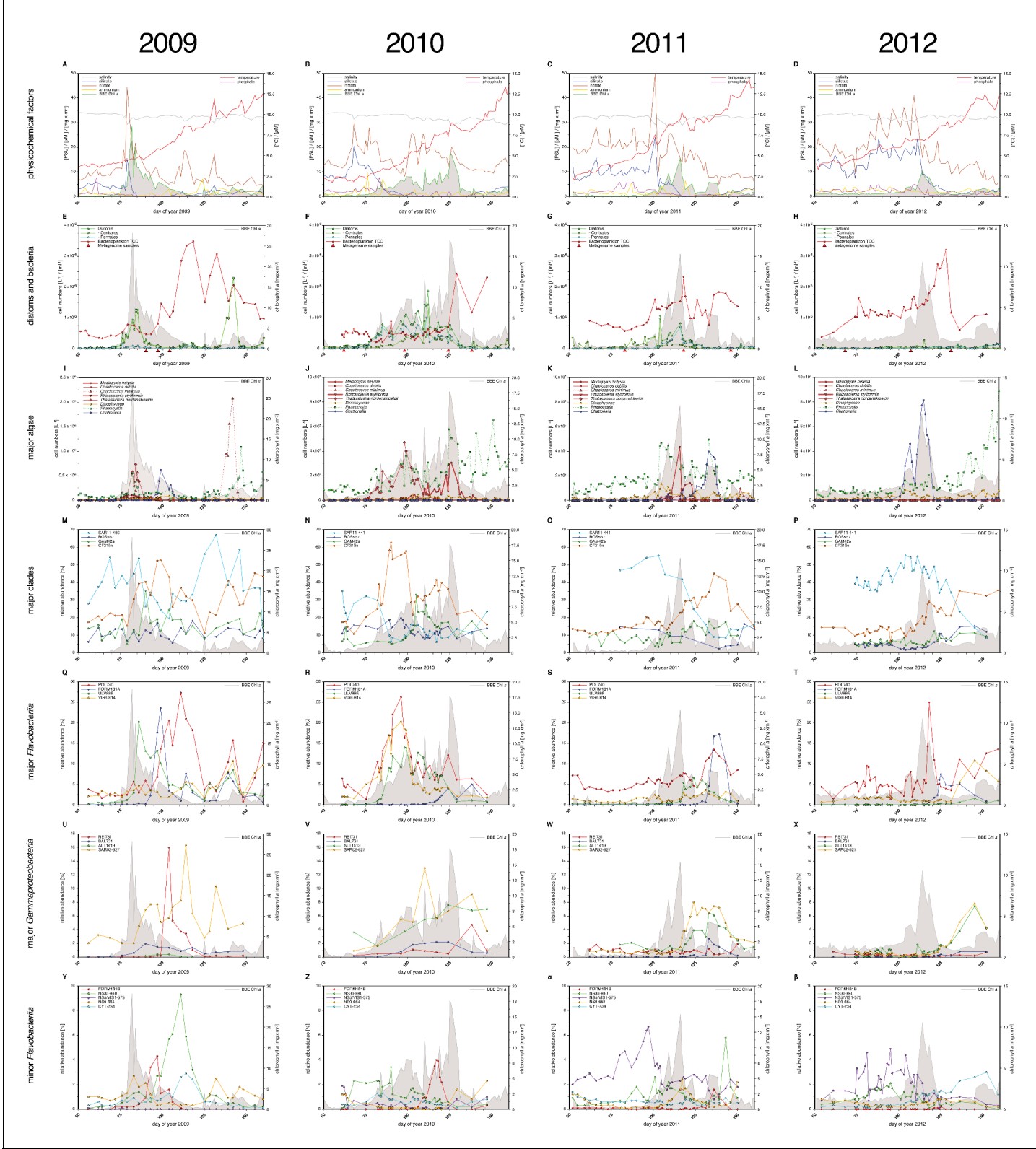

**Figure 3.** Physicochemical parameters, phytoplankton composition and bacterioplankton composition as assessed by CARD-FISH. *Sampling:* Surface seawater samples were taken at the North Sea island Helgoland between the main island and the minor island 'Düne' (station 'Kabeltonne', 54°11'03''N, 7°54'00''E) using small research vessels (http://www.awi.de/en/expedition/ships/more-ships.html) and processed in the laboratory of the Biological Station Helgoland within less than two hours after sampling. Cells for microscopic visualization methods were first fixed by the addition of

*Figure 3 continued on next page*

Figure 3 continued

formaldehyde to sampled seawater, which was then filtered directly onto 0.2 µm pore sized filters. *Physicochemical and phytoplankton data*: Physicochemical parameters and phytoplankton data were assessed in subsurface water on a weekday basis as part of the Helgoland Roads LTER time series as described in *Teeling et al. (2012)*. The Helgoland Roads time series is accessible via the public database Pangaea (http://www.pangaea.de) and can be used to assess long-term changes of the North Sea pelagic ecosystem. Left-hand side legends correspond to ordinates on the left, and right-hand side legends to ordinates on the right. A-D: Physicochemical measurements including measurements of BBE Chl *a* (chlorophyll *a* fluorescence by algal group analyzer sensor). Left ordinate: salinity [PSU], silicate [µM], nitrate [µM], ammonium [µM] and chlorophyll *a* [mg/m³]; right ordinate: temperature [°C] and phosphate [µM]. E-H: Counts of the diatom groups. I-L: Microscopic cell counts of the most abundant algae genera (red: diatoms; orange: dinoflagellates: green: haptophytes; blue: silicoflagellates). Algae with large cells and thus large biovolumes are depicted by bold solid lines and algae with small cells are represented by dotted lines. *Rhizosolenia styliformis* and *Mediopyxis helysia* feature large cells, whereas *Chaetoceros minimus* and in particular *Phaeocystis* species feature small cells. The latter typically reaches lengths of below 10 µm and *Phaeocystis* spp. biovolumes therefore typically are hundreds to thousands fold smaller than those of *R. styliformis* and *M. helyisa* cells (*Olenina, 2006*; *Loebl et al., 2013*). Physicochemical data are summarized in *Supplementary file 1*, and data on the major phytoplankton clades in *Supplementary file 2*. *Total cell counts and CARD-FISH of bacterioplankton*: E-H: TCC (total cell counts); red triangles depict sampling of metagenomes. M-β: Recurrent bacterioplankton clades as assessed by CARD-FISH (Catalyzed Reporter Deposition-Fluorescence in situ Hybridization) with the following probes: M-P (major bacterial groups): SAR11-486 and SAR11-441: alphaproteobacterial SAR11-clade; ROS537: alphaproteobacterial *Roseobacter* clade; GAM42a: *Gammaproteobacteria*; CF319a: *Bacteroidetes*. Q-T (major *Flavobacteriia* clades): POL740: genus *Polaribacter*; FORM181A: genus *Formosa*; ULV995: genus *Ulvibacter*; VIS6-814: genus-level clade VIS6 within the family *Cryomorphaceae-Owenweeksia*; U-X (major *Gammaproteobacteria* clades): REI731: genus *Reinekea*; BAL731: genus *Balneatrix*; ALT1413: families *Alteromonadaceae* and *Colwelliaceae*; SAR92-627: genus-level clade SAR92. Y-β (minor *Bacteroidetes* clades): FORM181B: species-specific for *Formosa* sp. Hel1_33_131; NS3a-840: NS3 marine group; NS5/VIS1-575: VIS1 genus-level clade within the NS5 marine group; NS9-664: NS9 marine group; CYT-734: *Cytophagia* clade *Marinoscillum*. Total and CARD-FISH cell counts are summarized in *Supplementary file 3* and the corresponding probes in *Supplementary file 4*.

---

*nordenskioeldii,* the silicoflagellate *Chattonella,* the haptophyte *Phaeocystis* and dinoflagellates dominated these blooms in terms of cell numbers (*Figure 3I–L*; *Supplementary file 2*). Relative abundances of these algae varied in no apparent order during the observed blooms, and we have yet to understand the factors that determine these variations. The sizes of the dominant algae taxa are different, with *Chaetoceros minimus* and in particular *Phaeocystis* spp. featuring the smallest cells and *Mediopyxis helysia* and *Rhizosolenia styliformis* featuring the largest cells. Spherical *Phaeocystis* spp. cells for example have estimated biovolumes of ~50 to 250 µm³, whereas elipsoid cylindrical *Mediopyxis helysia* cells have a biovolume of ~82,000 µm³ and for *Rhizosolenia styliformis* even a biovolume of ~282,000 µm³ has been reported (*Olenina, 2006*; *Loebl et al., 2013*). Considering biomass, the blooms were largely dominated by the diatoms *T. nordenskioeldii* and *M. helysia* and the silicoflagellate *Chattonella*. Blooms of these three algae were bimodal in all years with dominance of first *T. nordenskioeldii* followed by *Chattonella* in 2009 (*Figure 3I*), *T. nordenskioeldii* followed by *M. helysia* in 2010 (*Figure 3J*), *M. helysia* followed by *Chattonella* in 2011 (*Figure 3K*) and a pronounced bimodal bloom of *Chattonella* species in 2012 (*Figure 3L*). All blooms were accompanied by a notable decrease of silicate (*Figure 3A–D*; *Supplementary file 1*), which diatoms use for frustule formation (see *Yool and Tyrrell (2003)* for the controlling effect of silicate on diatom abundance).

Bloom maximum intensities decreased from 2009 to 2012 with chlorophyll *a* maxima (measured by fluorescence) reaching 28 mg m⁻³ (day 82), 18 mg m⁻³ (day 125), 15 mg m⁻³ (day 116), and 11 mg m⁻³ (day 114) in each respective year (*Figure 3A–D*; *Supplementary file 1*). Maximum bacterioplankton cell counts were observed either close to or after the chlorophyll maxima (*Supplementary file 3*). In 2009, the bacterioplankton peaked at $3.5 \times 10^6$ cells ml⁻¹ (day 118), 36 days after the Chl *a* maximum. In 2010, the peak in Chl *a* was broader and only four days ahead of the bacterioplankton peak abundance of $2.4 \times 10^6$ cells ml⁻¹ (day 129). In 2011, the Chl *a* peak was only two days ahead of the bacterioplankton peak abundance of $2.3 \times 10^6$ cells ml⁻¹ (day 118), whereas it was nine days ahead in 2012, where the bacterioplankton peaked at $3.2 \times 10^6$ cells ml⁻¹ (day 123).

## Bacterioplankton - diversity and bloom characteristics

DAPI and CARD-FISH cell staining (*Supplementary files 3* and *4*) showed that SAR11 dominated the bacterioplankton community in winter, but with the onset of each spring bloom relative abundances of *Bacteroidetes* followed by *Gammaproteobacteria* increased and finally surpassed those of the SAR11 (*Figure 3M–P*). *Bacteroidetes* reached higher maximum relative abundances than

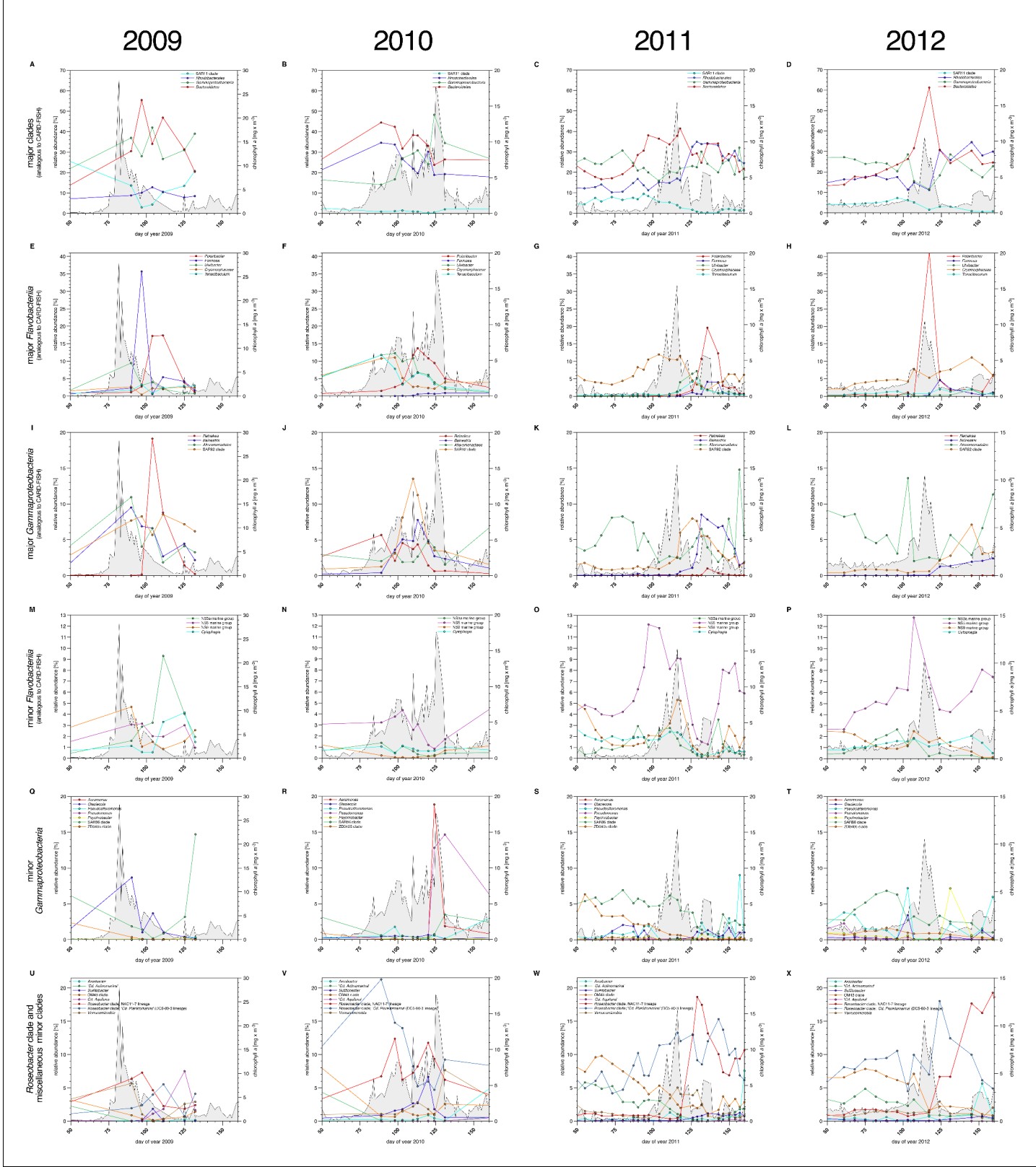

**Figure 4.** Bacterioplankton diversity as assessed by 16S rRNA gene tag sequencing. *Sampling*: Surface seawater samples were taken at the North Sea island Helgoland between the main island and the minor island 'Düne' (station 'Kabeltonne', 54°11'03''N, 7°54'00''E) using small research vessels (http://www.awi.de/en/expedition/ships/more-ships.html) and processed in the laboratory of the Biological Station Helgoland within less than two hours after

*Figure 4 continued on next page*

*Figure 4 continued*

sampling. Biomass of free-living bacteria for DNA extraction was harvested on 0.2 µm pore sized filters after pre-filtration with 10 µm and 3 µm pore sized filters to remove large debris and particle-associated bacteria. Biomass of the 0.2–3 µm bacterioplankton fraction was used for DNA extraction and subsequent 16S rRNA gene tag sequencing. *16S rRNA gene tag sequencing:* A total of 142 samples were collected for the years 2010 to 2012. After DNA extraction, the V4 region of the 16S rRNA gene was amplified using the primers 515F (5' GTGCCAGCMGCCGCGGTAA 3') and 806R (5' GGACTACHVGGGTWTCTAAT 3') (*Caporaso et al., 2011*). Sequencing was carried out on an Illumina (San Diego, CA, USA) MiSeq sequencer with and 2x250 bp chemistry. This dataset was complemented by 16S rRNA gene tags from 7 samples from our initial study on the 2009 spring bloom (*Teeling et al., 2012*). DNA of these samples was amplified using the primers 314F (5' CCTACGGGNGGCWGCAG 3') and 805R (5' GACTACHVGGGTATCTAATCC 3') (*Herlemann et al., 2011*) and sequenced on the 454 FLX Ti platform. *Data analysis:* All tag data were analyzed using the SILVAngs pipeline with the SILVA (*Quast et al., 2013*) v119 database. The SAR92 clade was subsequently reclassified to comply with the recently released SILVA v123, where the SAR92 no longer belong to the order *Alteromonadales*. The corresponding abundance data is summarized in *Supplementary file 5*. Time points from days 50 to 160 were plotted for all years. Panel A-P depict data that are analogous to the CARD-FISH data presented in *Figure 3*, with addition of the *Flavobacteriia* genus *Tenacibaculum* (E-H). Panels Q-X show minor *Gammaproteobacteria* clades (Q-T) and *Roseobacter* clades together with miscellaneous other minor clades (U-X) that were not tested by CARD-FISH probes.

*Gammaproteobacteria*, 40% (2012) to 60% (2010) as compared to 10% (2012) to 30% (2010), respectively.

*Bacteroidetes* genera *Polaribacter*, *Formosa*, and VIS6, a genus-level clade within the family *Cryomorphaceae* (*Gómez-Pereira et al., 2010*), peaked each year with relative abundances well above 5%, reaching relative abundances of up to 25% sometimes within less than a week (*Figure 3Q–T*). *Ulvibacter* reached similar peak abundances with the exception of 2012, where this genus never surpassed 2% and ranged below 1% most of the time. Within *Gammaproteobacteria* the genus-level SAR92 clade responded notably in all years increasing from background levels below 1% to relative abundance of 8% to 16%. Members of the *Alteromonadales* families *Alteromonadaceae* and *Colwelliaceae* bloomed in three (2010: 8%; 2011: 6%; 2012: 7%), and genus *Reinekea* in two (2009: 16%; 2010: 5%) of the years (*Figure 3U–X*). Some less abundant, but nevertheless recurrent taxa included the genus *Balneatrix* within *Gammaproteobacteria* with relative abundances up to 2% (*Figure 3U–X*). Minor recurring groups of *Bacteroidetes* (*Figure 3Y-β*) included the NS3a marine group (9% in 2009 and 6% in 2011), the genus-level VIS1 clade within the NS5 marine group detected before the Chl *a* peaks of 2011 (7%) and 2012 (5%), and the *Cytophagia* clade *Marinoscillum* that reached 1–3% abundance most years after initial blooms.

We used complementary 16S rRNA gene tag sequencing for the detection of bacterioplankton clades that were not recovered by CARD-FISH probes (*Supplementary file 5*). Relative proportions of 16S tags from distinct clades correlated for the most part those inferred from CARD-FISH cell counts (*Figure 4A–P*), but members of SAR11 were substantially underreported - a known limitation of the 806R primer used in V4 amplification for 2010 to 2012 samples (*Apprill et al., 2015*). Additional abundant clades detected in the 16S amplicon data comprised the *Flavobacteriia* genus *Tenacibaculum* (*Figure 4U–X*) that bloomed in 2010 (read frequencies of max. ~12%) and 2011 (max. ~5%). Within *Gammaproteobacteria*, clades with read frequencies ≥5% in at least one year comprised the genera *Aeromonas*, *Glaciecola*, *Pseudoalteromonas*, *Pseudomonas*, *Psychrobacter* and the SAR86 and ZD0405 clades (*Figure 4Q–T*). Within the alphaproteobacterial *Rhodobacteriaceae*, high abundances of 'Candidatus Planktomarina temperata' (DC5-80-3 lineage) and the NAC11-7 clade were detected reaching 6–21% and 7–19% of the tag data, respectively (*Figure 4U–X*). Also within *Alphaproteobacteria* the genus *Sulfitobacter* peaked with a read frequency of ~7% in 2010 (*Figure 4V*), and within *Betaproteobacteria* the order *Methylophilales* (dominated by OM43 clade members) was detected with high relative abundances of up to ~10% before blooms, which decreased with bloom progression. *Verrucomicrobia* (*Figure 4U–X*) were detected with decreasing peak read frequencies of 7.7% (2010), 5.0% (2011) and 2.9% (2012). This decrease corresponds to decreasing bloom intensities, which supports a proposed role of *Verrucomicrobia* in polysaccharide decomposition (e.g. *Martinez-Garcia et al., 2012*).

Within *Bacteroidetes*, *Gammaproteobacteria* and *Rhodobacterales* a total of eleven clades peaked during at least two of the four spring blooms with relative cell abundances or, for those clades that were not assessed by CARD-FISH, relative read frequencies ≥5%. These were six *Flavobacteriia* clades (*Formosa*, *Polaribacter*, NS3a marine group, *Tenacibaculum*, *Ulvibacter*, VIS6 clade

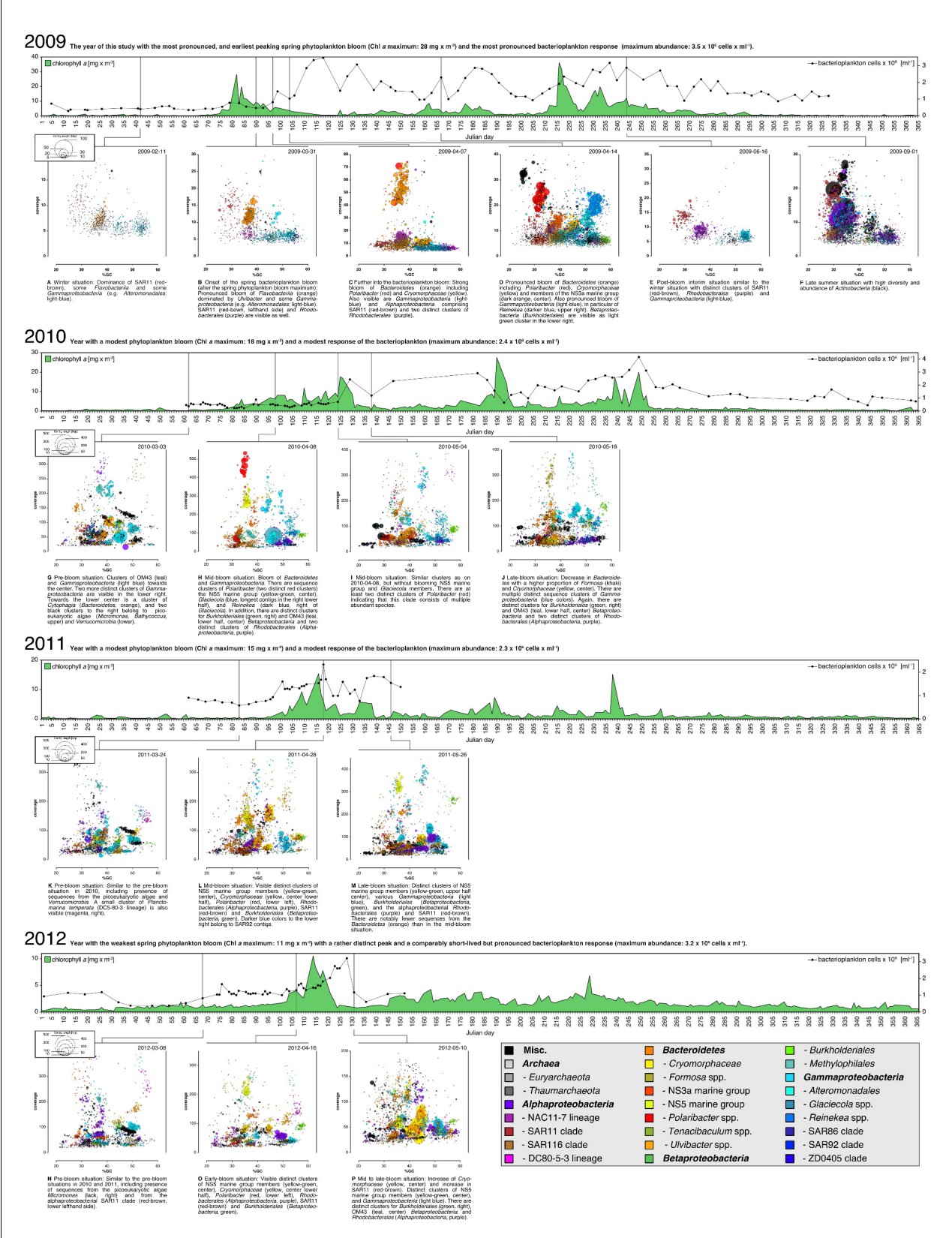

**Figure 5.** Taxonomic classification of bacterioplankton metagenomes *Sampling*: Surface seawater samples were taken at the North Sea island Helgoland between the main island and the minor island 'Düne' (station 'Kabeltonne', 54°11.03' N, 7°54.00' E) and processed in the laboratory of the

*Figure 5 continued*

Biological Station Helgoland within less than two hours after sampling. Biomass of free-living bacteria was harvested on 0.2 µm pore sized filters after pre-filtration with 10 µm and 3 µm pore sized filters to remove large debris and particle-associated bacteria. *Sequencing:* Community DNA was extracted and sequenced; 2009 samples were sequenced on the 454 FLX Ti platform, and 2010-2012 samples on the Illumina HiSeq2000 platform (16 metagenomes in total). Reads were assembled using Newbler (2009) or a combination of SOAPdenovo and Newbler (2010-2012) and the resulting contigs were taxonomically classified (*Supplementary file 9*). *Visualization:* The resulting metagenome contigs are visualized as bubbles with radii that are proportional to their lengths and colors that indicate their predicted taxomomic affiliations. These bubbles are drawn in planes that are defined by the contig's GC contents and coverage values. Colors are restricted to selected abundant taxa (see legend below) to highlight distinct clusters, mostly from the *Bacteroidetes, Alphaproteobacteria, Betaproteobacteria* and *Gammaproteobacteria*. Likewise only contigs are shown that exceed a minimum length of 2750 bp for pyrosequencing data (2009) and 15,000 bp for Ilumina data (2010-2012), respectively. Sparse contigs with very high coverage or GC contents below 20% or above 60% were also excluded from visualizations. The 16 metagenomes are shown arranged in order on yearly timescales that depict chlorophyll *a* contents as proxies for phytoplankton abundance. *Metagenome sizes\*:* 2009-02-11: 49.1 Mbp / 2009-03-31: 44.9 Mbp / 2009-04-07: 52.7 Mbp / 2009-04-14: 96.0 Mbp / 2009-06-16: 29.8 Mbp / 2009-09-01: 79.2 Mbp 2010-03-03: 537.3 Mbp / 2010-04-08: 325.8 Mbp / 2010-05-04: 453.0 Mbp / 2010-05-18: 512.3 Mbp 2011-03-24: 629.1 Mbp / 2011-04-28: 541.8 Mbp / 2011-05-26: 604.0 Mbp 2012-03-08: 574.0 Mbp / 2012-04-16: 543.9 Mbp / 2012-05-10: 614.1 Mbp \*sums of assembled bases

*Cryomorphaceae*), three *Gammaproteobacteria* clades (*Alteromonadaceae/Colwelliaceae, Reinekea* and SAR92), and two *Roseobacter* clades (DC5-80-3 and NAC11-7).

Each year a succession was observed within the *Flavobacteriia* and *Gammaproteobacteria* clades. The succession in the *Flavobacteriia* was more pronounced than in the *Gammaproteobacteria*, but the sequence of clades varied. Spearman rank correlation analyses revealed that the abundances of the most prominent *Flavobacteriia* clades were for the most part correlated with multiple algae groups and physiochemical factors (*Supplementary file 6*). According to linear regression analyses, the strongest abiotic predictors were temperature, salinity, silicate and nitrate, and the strongest biotic predictors were *Phaeocystis* spp. haptophytes, *Rhizosolenia* spp., *Chaetoceros debilis*, and *Chaetoceros minimus* diatoms and the silicoflagellate *Chattonella* (*Supplementary file 7*). It should be noted though that linear regressions were computed based on log-transformed abundance data and not algae volumes (which were not measured). Thus, the influence of the rather small cell-sized algae such as *Chaetoceros minimus* is likely overestimated. Such limitations notwithstanding it is noteworthy that in no case a simple one-to-one relationship between specific algae and specific bacterioplankton groups was detected. The strongest significant ($p < 0.05$) correlations were obtained for the *Ulvibacter* clade that was positively correlated with diatoms and haptophytes and negatively correlated with silicoflagellates. Further results comprised an opposite trend for the VIS1 clade of the NS5 marine group, and a correlation of *Polaribacter* and *Chattonella* abundances (see *Supplementary file 6* for details).

## Bacterioplankton - genetic repertoires

In total, 16 metagenomes of free-living bacterioplankton (0.2–3 µm) were generated from time points before, during and after spring phytoplankton blooms, six during 2009 using the 454 FLX Ti platform that were published previously (*Teeling et al., 2012*) and ten during 2010–2012 using the Illumina HiSeq2000 platform. Most of the 454 (0.5–4 pico titer plates / metagenome) and all of the Illumina metagenomes (1 lane / metagenome; 2x150 bp) were deeply sequenced (*Supplementary file 8*) with final assembled contigs of up to 96 kbp and 458 kbp, respectively.

Taxonomic classification of the metagenome contigs resulted in identification of major bloom-associated clade sequence bins (*Supplementary file 9*), including *Formosa, Polaribacter*, the NS3a and NS5 marine groups, and *Cryomorphaceae* of the *Flavobacteriia* and *Alteromonadales, Reinekea, Glaciecola* and the SAR92 clade of the *Gammaproteobacteria*. Classification was poor, however, for *Ulvibacter (Flavobacteriia)* and *Balneatrix (Gammaproteobacteria)*, most likely since the only available reference genomes (unidentified eubacterium SCB49; *Balneatrix alpica*) were too distant from North Sea representatives. Clone libraries from 2009 (*Teeling et al., 2012*) indicated 16S rRNA similarities of only 94% and 91%, respectively, for these two clades. Other abundant clades comprised the betaproteobacterial *Burkholderiales* and *Methylophilales* (including the OM43 clade), the alphaproteobacterial SAR116 and *Roseobacter* NAC11-7 clade, and the gammaproteobacterial SAR86 and ZD0405 clades. Lower abundant clades comprised, amongst others, the OM60 (NOR5) group, the AEGEAN-169 group, and *Sulfitobacter*.

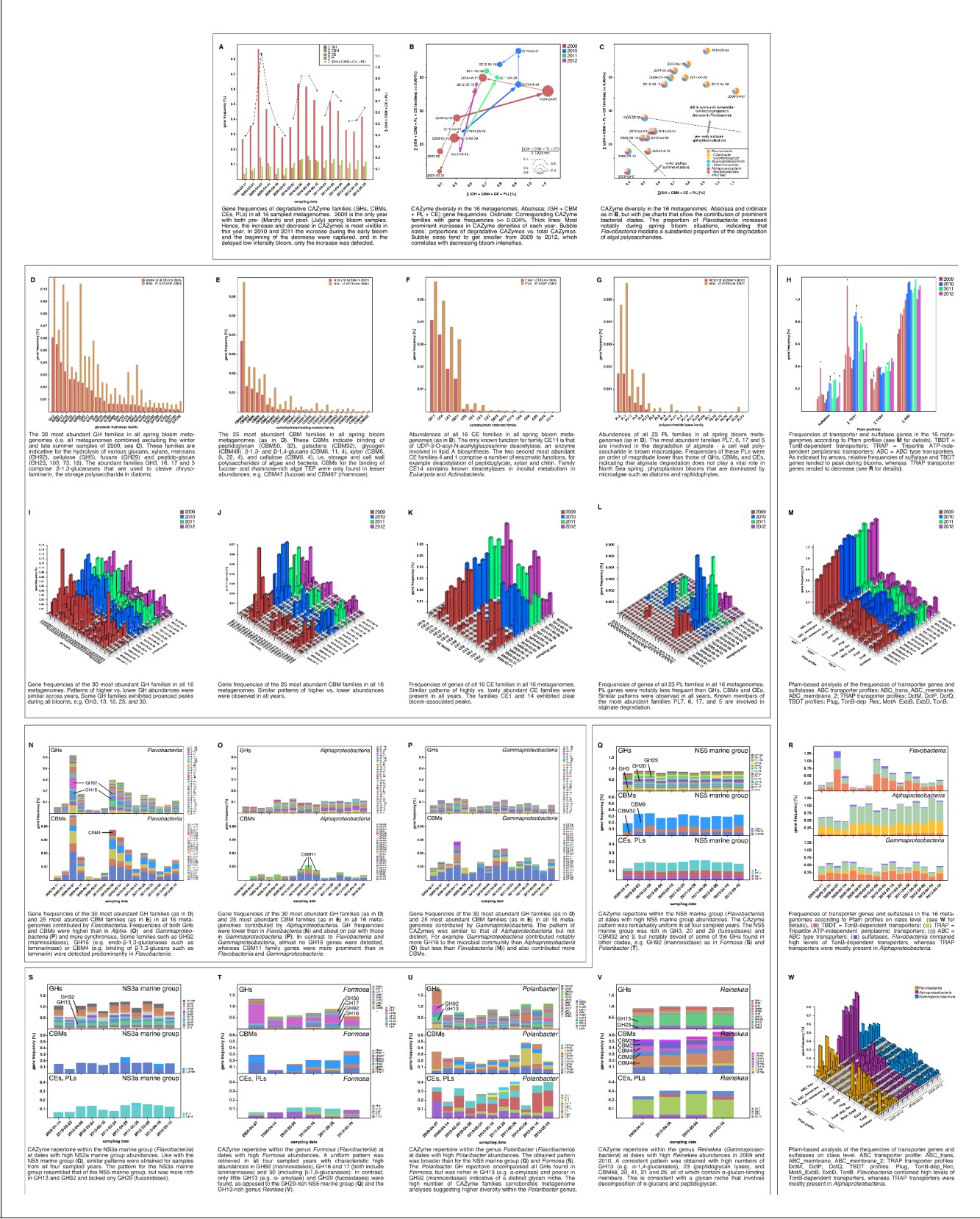

**Figure 6.** Metagenome functional analyses: CAZyme, sulfatase and transporter gene frequencies. *Sampling:* Surface seawater samples were taken at the North Sea island Helgoland between the main island and the minor island Düne' (station 'Kabeltonne', 54°11′03′′N, 7°54′00′′E) and processed in the

*Figure 6 continued*

laboratory of the Biological Station Helgoland within less than two hours after sampling. Biomass of free-living bacteria was harvested on 0.2 μm pore sized filters after pre-filtration with 10 μm and 3 μm pore sized filters to remove large debris and particle-associated bacteria. *Sequencing:* Community DNA was extracted and sequenced. 2009 samples were sequenced on the 454 FLX Ti platform, and 2010–2012 samples on the Illumina HiSeq2000 platform (16 metagenomes in total). Reads were assembled using Newbler (2009) or a combination of SOAPdenovo and Newbler (2010–2012) and the resulting contigs were taxonomically classified (*Supplementary file 9*). *Metagenome sizes\*:* 2009-02-11: 49.1 Mbp / 2009-03-31: 44.9 Mbp / 2009-04-07: 52.7 Mbp / 2009-04-14: 96.0 Mbp / 2009-06-16: 29.8 Mbp / 2009-09-01: 79.2 Mbp 2010-03-03: 537.3 Mbp / 2010-04-08: 325.8 Mbp / 2010-05-04: 453.0 Mbp / 2010-05-18: 512.3 Mbp 2011-03-24: 629.1 Mbp / 2011-04-28: 541.8 Mbp / 2011-05-26: 604.0 Mbp 2012-03-08: 574.0 Mbp / 2012-04-16: 543.9 Mbp / 2012-05-10: 614.1 Mbp \*sums of assembled bases *Data Analysis:* CAZymes were predicted as consensus of searches against the CAZy, dbCAN and Pfam databases with custom E-value cutoffs (*Supplementary file 11*). Sulfatase and transporter genes were predicted based on HMMER searches against the Pfam databases with an E-value cutoff of E-5. Gene frequencies were computed as [(sum of average coverage of target genes) \*100 / (sum of average coverage of all genes)]. All dates in the graphs are in the format [yyyy-mm-dd].

The following figure supplements are available for figure 6:

**Figure supplement 1.** CAZyme repertoire within the family *Cryomorphaceae (Flavobacteriia)* at dates with high *Cryomorphaceae* abundances.

**Figure supplement 2.** CAZyme repertoire within the order *Alteromonadales (Gammaproteobacteria)* at dates with high abundances of *Alteromonadales*.

We plotted contig GC contents versus coverage to evaluate our taxonomic classification, which in some cases allowed to assess the coherence of some of the clades (*Figure 5*). For example, *Reinekea* (*Figure 5D,H,I*) and the NS5 marine group (*Figure 5H,L,M,O*) were mostly represented by distinct clusters, whereas *Polaribacter* (*Figure 5D,H,I*) was almost always represented by at least two clusters indicating the presence of sub-populations. In general, the number of clusters increased from pre-bloom to mid-bloom situations and decreased slightly towards late bloom situations and notably towards post-bloom situations. This tendency was more evident in 2009, the year with the highest bloom intensity and the largest number of metagenome samples spanning a broader time-span (*Figure 5A–F*). It is noteworthy that high in situ abundance did not always correlate with good metagenome assemblies. SAR11 for example, while highly abundant in all metagenome datasets, yielded few large contigs, possibly due to population heterogeneity and presence of hyper-variable regions described in sequenced SAR11 genomes (*Wilhelm et al., 2007*).

Analyses of functional genes identified in the bacterioplankton metagenomes revealed that increases in *Flavobacteriia* relative abundance during blooms was always accompanied by an increase in community-wide CAZyme gene frequency as well as an increase in the diversity of CAZyme families (*Figure 6A–C*). As blooms subsided, CAZyme frequencies also declined. This was most pronounced for more densely sampled years in 2009 and 2010. In 2012 the decline in CAZymes was not captured as the last metagenome sample was taken before the bloom decline (*Figure 6A*, x-axis).

The 20 glycoside hydrolase families with the highest mean abundances during bloom dates were, in descending order, GH3, 23, 13, 16, 103, 73, 92, 2, 17, 30, 5, 20, 36, 65, 29, 1, 42, 31, 81, and 18 (*Figure 6D*). While most of these families comprise a diverse range of functions, they are indicative for the hydrolysis of certain glucans, xylans, mannans (GH92), cellulose (GH5), fucans (GH29), and peptidoglycan (GH23, 103, 73, and 18). The abundant families GH3, 16, 17 and 5 comprise β-1,3-glucanases and the family GH30 β-1,6-glucanases. Both enzymes are involved in the cleavage of chrysolaminarin. Chrysolaminarin, a mostly linear β-1,3-glucan with occasional β-1,6 branches, is the storage polysaccharide in diatoms and thus one of the most abundant polysaccharides on Earth.

The ten families of carbohydrate-binding modules with the highest mean abundances during blooms were, in descending order, CBM50, 32, 6, 9, 48, 11, 22, 57, 20, and 4 (*Figure 6E*). These CBM domains are indicative for the binding of peptidoglycan (CBM50), galactans (CBM32), glycogen (CBM48), β-1,3- and β-1,4-glucans (CBM6, 11, 4), xylan (CBM6, 9, 22, 4), and cellulose (CBM6, 4). This suggests a pronounced specialization of the bacterial community in the acquisition of storage polysaccharides (β-1,3-glucans such as chrysolaminarin; α-1,4-glucans such as starch and glycogen) and cell wall polysaccharides (xylose, cellulose and peptidoglycan) of both algae and bacteria. By contrast, CBMs for the binding of algal TEP, which consists predominantly of fucose- and

rhamnose-rich anionic sulfated heteropolysaccharides (*Passow, 2002*), were only found in lower abundances, e.g. CBM47 (fucose) and CBM67 (rhamnose).

Among carbohydrate esterase families, CE11, 4, 1, and 14 exhibited the highest mean abundances during blooms (*Figure 6F*). The only known function for family CE11 is that of the UDP-3-O-acyl-N-acetylglucosamine deacetylase, an enzyme involved in lipid A biosynthesis. The second and third most abundant CE families 4 and 1 comprise a number of enzymatic functions including deacetylation of peptidoglycan, xylan, and chitin.

Finally, the polysaccharide lyase families PL6, 7, 17 and 5 constituted the most abundant PL families during bloom dates (*Figure 6G*). Known members of these PL families are all involved in the usage of alginate. However, gene frequencies of these PL families were an order of magnitude below those of abundant GHs, CBMs, and CEs, indicating that alginate degradation does not play a vital role in North Sea spring phytoplankton blooms. Alginate is a cell wall constituent in brown macroalgae, however, the microalgae that dominate North Sea spring blooms are devoid of alginate.

For all of these GH, CBM, PL, and CE families, we observed remarkably similar gene frequency patterns during all four spring blooms, often with peaks in the same families (*Figure 6I–L*). Alongside CAZymes, sulfatase and TonB-dependent transporter (TBDT) gene frequencies also increased during blooms while tripartite ATP-independent periplasmic (TRAP) transporter genes showed an almost opposite trend (*Figure 6H,M*). A class-level analysis revealed that sulfatases and TBDT genes were predominantly present in *Bacteroidetes*, whereas TRAP transporters were mostly present in *Alphaproteobacteria* (*Figure 6R,W*). The observed shifts in sulfatase, TBDT and TRAP transporter frequencies hence reflect the shifts in relative abundance between these two classes. This is in agreement with our study on the 2009 spring bloom (*Teeling et al., 2012*) and furthermore demonstrates recurrence of this phenomenon during four consecutive years.

Class-level analyses of the most abundant GH and CBM families (*Figure 6N–P*) showed that *Flavobacteriia* not only contributed more total CAZymes to the microbial community than *Alpha*- and *Gammaproteobacteria*, but also exhibited a tighter coupling between GH and CBM genes with highly similar abundance profiles (*Figure 6N*). The distribution of families was also more uneven in *Flavobacteriia*, indicative of a more pronounced substrate specialization compared to *Alpha*- and *Gammaproteobacteria*. GH92 (mannosidase) and GH16 (including β-1,3-glucanase) genes, for example, were predominantly found in *Flavobacteriia*, possibly indicating more readiness to decompose mannans and chrysolaminarin.

Metagenome taxonomic classification provided sufficient data for analysis of CAZyme repertoires of the flavobacterial NS5 marine group, NS3a marine group, *Formosa, Polaribacter,* and *Cryomorphaceae*, and the gammaproteobacterial *Alteromonadales* and *Reinekea* clades (*Supplementary file 10*). For most of these clades, the analyses revealed fingerprint-like patterns, which corroborates the hypothesis that these clades have distinct glycan niches that are relatively stable across years. For example, the NS5 marine group (*Figure 6Q*) was rich in GH3, 20 and 29 (fucosidases), but notably devoid of GH92 (mannosidases). By contrast, *Formosa* and *Polaribacter* clades (*Figure 6T,U*) contained higher abundances of GH92 genes. The *Formosa* CAZyme profile was also characterized by high proportions of GH16, 17, and 30 families, which all contain enzymes that can decompose chrysolaminarin. *Polaribacter* contained a broader set of CAZymes that included all families found in *Formosa*, however, *Polaribacter* was richer in GH13 (e.g. α-amylase) and poorer in GH92 (mannosidases) than *Formosa*. Likewise, the GH repertoires of the NS3a and NS5 marine groups were similar (*Figure 6Q,S*), but the NS3a marine group was richer in GH13 and 32 and devoid of GH29 family fucosidases. The high number of CAZyme families in *Polaribacter* corroborated metagenome bin analyses that suggested a higher diversity within this clade. CAZyme gene frequencies were much lower in the *Cryomorphaceae* than in the other investigated *Flavobacteriia* clades with GH frequencies barely exceeding 0.5% (*Figure 6—figure supplement 1*). This suggests a different ecophysiological niche and a distinct role of the *Cryomorphaceae* during phytoplankton blooms.

For *Gammaproteobacteria*, recurring patterns were detected for the prominent *Alteromonadales* and *Reinekea* clades. *Alteromonadales* contained some of the GH families that play important roles during phytoplankton blooms, such as GH13 and 16, but were notably poor in or even devoid of others, such as GH29 and GH92, respectively (*Figure 6—figure supplement 2*). In contrast to other prominent clades, we did not obtain sufficient metagenome sequences for *Reinekea* for all four years, but only for 2009 and 2010 (*Figure 6V*). However, the *Reinekea* CAZyme patterns of 2009 and 2010 were well conserved with high proportions of GH23 and 13, and CBM48, 20, 41, 21, and

25. The GH23 family comprises peptidoglycan lyases and the GH13 family contains α-1,4-glucanases (e.g. α-amylase). CBM48, 20, 41, 21, and 25 all bind α-1,4-glucans such as starch and glycogen, and the ubiquitous CBM50 contains peptidoglycan-binding members. These results are consistent with a glycan niche that involves decomposition of external α-1,4-glucans and possibly peptidoglycan.

## Discussion

Nutrient-poor marine surface waters are dominated by clades such as SAR11 and *Prochlorococcus*. Both feature small, reduced genomes and can use sunlight and small organic molecules (e.g. *Gómez-Pereira et al., 2013*). The otherwise heterotrophic SAR11 use proteorhodopsin for supplemental phototrophy (*Giovannoni et al., 2005*) and phototrophic *Prochlorococcus* cyanobacteria are capable of supplemental uptake of amino acids (*Zubkov et al., 2003*) and glucose (*Muñoz-Marín et al., 2013*). Microbial communities in nutrient-rich 'green' surface oceans by contrast feature higher proportions of heterotrophic species that feed on more complex organic substrates. In particular during phytoplankton blooms, the release of algae-derived organic matter selects for fast growing species with genomic adaptations towards algal biomass remineralization. These are typically members of the *Flavobacteriia* and *Gammaproteobacteria* classes and the alphaproteobacterial *Roseobacter* clade. Similarly adapted species from these clades compete for substrates during phytoplankton blooms with variation in which species prevail. Despite this stochastic effect, the most well-adapted species will be successful more often and thus exhibit patterns of annual recurrence.

During spring phytoplankton blooms at Helgoland in the North Sea, we observed recurrent bloom-associated abundance peaks of in particular flavobacterial clades, namely *Formosa*, *Polaribacter*, the NS3a marine group, *Tenacibaculum*, *Ulvibacter*, and the *Cryomorphaceae* VIS6 clade. Within *Gammaproteobacteria Alteromonadaceae/Colwelliaceae*, *Reinekea*, and the SAR92 clade were clearly bloom-associated and recurrent as was *Methylophilales* within *Betaproteobacteria*, and 'Cd. *Planktomarina temperata*' from the DC5-80-3 lineage (a.k.a *Roseobacter* clade affiliated = RCA group) and the NAC11-7 lineage within the *Roseobacter* clade. It has already been shown that the abundant North Sea isolate 'Cd. *Planktomarina temperata*' RCA23[T] is associated with decaying phytoplankton (*Giebel et al., 2011*) and high abundances and in particular high activity have been reported during a spring phytoplankton bloom event in the North Sea of 2010 (*Voget et al., 2015*; *Wemheuer et al., 2015*). High activities of members of the RCA and the SAR92 clades during North Sea spring phytoplankton blooms have also been reported in 2009 (*Klindworth et al., 2014*) and 2010 (*Wemheuer et al., 2015*), just as an increase of *Bacteroidetes* of the genera *Marinoscillum and Polaribacter* during 2010 (*Wemheuer et al., 2015*).

Many of the other clades we report here (including some low abundance groups) have been found during blooms of dinoflagellates, including AEGEAN-169, *Alteromonadales*, NS3a marine group, NS5 marine group, OM43, OM60 (NOR5), SAR116, SAR86, and ZD0405 (*Yang et al., 2015*) or *Cryomorphaceae*, *Glaciecola* and *Sulfitobacter* (*Tan et al., 2015*). The OM43 clade (order *Methylophilales*) comprises methylotrophs known to feed on algae C1 compounds (*Halsey et al., 2012*), and it has been reported that *Sulfitobacter* species SA11 shares a mutually beneficial exchange of compounds with the diatom species *Pseudo-nitzschia multiseries* (*Amin et al., 2015*).

There is of course unaddressed diversity in all these clades. Some of the genera might be dominated by a single species while others might be more diverse with considerable variation in competitive success between bloom events. Nevertheless, the high level of recurrence in particular of flavobacterial clades indicates a strong selection of few clades highly adapted for the manipulation and uptake of specific and complex polysaccharides (and likely other biopolymers) and disagrees with substantial levels of sloppy-feeding by these bacteria that would allow other less adapted clades to arbitrarily reach high abundances via cross-feeding. This might be attributed to the capability of *Bacteroidetes* for very efficient macromolecule uptake as it has been recently shown for uptake of α-mannan by the human gut bacterium *Bacteroides thetaiotaomicron* (*Cuskin et al., 2015*). This bacterium binds α-mannan macromolecules to its surface, followed by rapid cleavage into larger oligomers that are immediately imported via a TonB-dependent transporter (TBDT) into the periplasm without detectable loss. The bulk of the degradation into smaller molecules takes place in the periplasm where the substrate is secure from outside competitors before transport into the cytoplasm.

TonB-dependent transporters are not specific to *Bacteroidetes*, but it seems that only *Bacteroidetes* have evolved a functional coupling of SusC-like TBDT porins with SusD-like TonB-dependent receptors (TBDRs) that bind and guide the substrate to the porin. At least so far, only *Bacteroidetes* genomes feature characteristic *susCD* gene tandems. Within *Bacteroidetes* genomes such tandems are frequently found in so-called polysaccharide utilization loci (PULs; [*Sonnenburg et al., 2010*]). PULs are operons or regulons where one or more *susCD* gene tandems are co-located with CAZymes. Further frequent accessory genes in PULs encompass transcriptional regulators, proteases, transporter components and sulfatases. The latter are required for the desulfation of sulfated polysaccharides, which marine algae produce in large quantities. The diversity of PULs in marine *Bacteroidetes* genomes is high and largely unexplored and so far only few PULs have been linked to dedicated algal polysaccharides (e.g. *Hehemann et al., 2012*; *Kabisch et al., 2014*; *Xing et al., 2015*). The large, complex and efficient PUL uptake systems might explain why *Flavobacteria* consistently outcompeted *Gammaproteobacteria* during the onset of all blooms.

It is noteworthy that we observed a shift in bacterioplankton biodiversity alongside a shift in functional gene repertoires for the major clades in all four years of this study. The abundance of CAZymes and sulfatases increased from pre- to mid-bloom situations and leveled off post-bloom. Likewise, similar abundance patterns were observed for the most abundant CAZyme families in all studied years.

We did observe an increase in the abundance of TBDRs during bloom situations, and we have shown previously that TBDRs are among the most abundantly expressed proteins during the bacterial mineralization of algae biomass (*Teeling et al., 2012*). The relevance of TBDRs in nutrient-rich oceanic regions has been also supported by in situ metaproteome studies of samples from the South Atlantic Ocean (in particular at coastal upwelling zones; [*Morris et al., 2010*]) and from the Antarctic Southern Ocean (*Williams et al., 2013*).

The recurrent patterns in bacterioplankton diversity and functional repertoires during the four studied spring blooms are remarkable in view of the variation among algae taxa. For example, even though algal biomass of the 2012 spring bloom was dominated by silicoflatellate *Chattonella* spp. and not by diatoms as was true for the three preceding years, the respective bacterioplankton communities were strikingly similar. Likewise only few bacterioplankton taxa seemed to be weakly correlated with dedicated phytoplankton taxa, and no clear correlation was found between distinct bacterioplankton taxa and individual distinct diatom clades in statistical analyses (*Supplementary files 6* and *7*). It seems that phytoplankton community composition did not exert a strong effect on the composition of the free-living non-phycosphere bacterioplankton community. Instead, the dominating algae (in terms of biomass) seemed to produce similar or perhaps identical types of substrates for specifically adapted clades of heterotrophic bacterioplankton. It is therefore conceivable that recurrence is more pronounced on the functional level than on the taxonomic level, since species from different taxa with similar ecophysiological niches might functionally substitute each other in different years. This hypothesis is supported by the bacterioplankton communities' similar CAZyme gene repertoires in the 2009 to 2012 spring blooms and in particular the consistency on class level that was almost unaffected by distinct blooming clades, yet needs to be further tested by deep metatranscriptome sequencing and metaproteomics during multiple spring phytoplankton blooms in future studies. Considering the extent of recurrence, our combined metagenome data (>5 million predicted proteins) should provide sufficient search space for such an analysis.

The existence of recurrent key players during North Sea spring phytoplankton blooms suggests that the bacterioplankton community composition during and after such blooms is governed by deterministic effects. We have shown before that temperature exerts a strong effect on North Sea bacterioplankton as it selects for temperature-dependent guilds, for example when comparing spring and summer blooms (*Lucas et al., 2015*). Within short-lived spring blooms, however, the supply of algae-derived organic matter is among the main factors that shape the bacterioplankton composition. In particular the different types of structurally distinct polysaccharides that algae produce in large quantities seem to exert such substrate-induced forcing. Since *Flavobacteriia* are more specialized on polysaccharides than *Gammaproteobacteria*, this would also explain, why *Flavobacteriia* dominated the recurrent clades (*Figure 3Q–T*, *Figure 4E–J*) and *Gammaproteobacteria* clades exhibited more stochastic peaks (*Figure 4Q–T*).

At the beginning of a bloom, most available polysaccharides will be exopolysaccharides, but as the bloom commences and algae become senescent, more and more cellular algal substrates are

released, culminating in the bloom's final die off phase. Bacteria will naturally consume the more degradable substrates such as storage polysaccharides (e.g. chrysolaminarin) first, and more recalcitrant substrates (e.g. branched and sulfated polysaccharides) later. TEP for example seems to undergo such selective feeding, as it has been suggested that in particular fucose-rich TEP is less readily degraded than mannose and galactose rich TEP (for review see *Passow, 2002*). Such selective feeding creates an additional change in substrate availability and leads to a succession of substrate niches for specifically adapted bacterioplankton clades to grow.

## Concluding remarks

Bacterioplankton communities during spring phytoplankton blooms in the coastal North Sea undergo swift and dynamic composition changes and thus are difficult to investigate. Nonetheless, we found clades that recurrently reached high abundances among *Flavobacteriia* (*Formosa, Polaribacter*, NS3a marine group, *Ulvibacter*, VIS6 clade *Cryomorphaceae, Tenacibaculum*), *Gammaproteobacteria* (*Alteromonadaceae/Colwelliaceae*, SAR92, *Reinekea*) and *Roseobacter* clade *Alphaproteobacteria* (DC5-80-3, NAC11-7). Recurrence was not only detectable on the taxonomic but also on the functional level with a highly predictable increase in TonB-dependent polysaccharide uptake systems and distinct CAZyme patterns. The niches of abundant bacterioplankton clades are more complex and manifold than the glycan niches that we explore in this study. CAZymes, however, have the advantage that they allow linking of gene repertoires and possible environmental functions in a way currently not feasible for other macromolecules such as proteins and lipids. Our results suggest that besides stochastic also deterministic effects influence phytoplankton-bacterioplankton coupling during blooms. They indicate that during spring phytoplankton blooms similar principles of resource partitioning and specialization are at play as within human gut microbiota that decompose fiber-rich plant material, albeit at a much larger scale. Rather the availability of substrates commonly occurring in microalgae than one-to-one interactions of particular phytoplankton and bacterioplankton species caused the succession of free-living bacterioplankton clades.

# Materials and methods

## Phytoplankton and physicochemical data

Physicochemical parameters (*Supplementary file 1*) and phytoplankton data (*Supplementary file 2*) were assessed in subsurface water on a weekday basis as part of the Helgoland Roads LTER time series. Details on the acquisition of these data have been described previously (*Teeling et al., 2012*). The Helgoland Roads time series is accessible via the public database Pangaea (http://www.pangaea.de).

## Bacterioplankton

Sampling of bacterioplankton was carried out as described previously (*Teeling et al., 2012*). In brief, surface seawater samples were taken at the long-term ecological research station 'Kabeltonne' (54° 11.3' N, 7° 54.0' E) at the North Sea island Helgoland using small research vessels (http://www.awi.de/en/expedition/ships/more-ships.html) and processed in the laboratory of the Biological Station Helgoland within less than two hours after sampling.

Biomass of free-living bacteria for DNA extraction was harvested on 0.2 µm pore sized filters after pre-filtration with 10 µm and 3 µm pore sized filters to remove large debris and particle-associated bacteria. By contrast, cells for microscopic visualization methods were first fixed by the addition of formaldehyde to sampled seawater, which was then filtered directly onto 0.2 µm pore sized filters. All filters were stored at -80°C until further use.

## Microscopy: total cell counts, CARD-FISH

Assessment of absolute cell numbers and bacterioplankton community composition was carried out as described previously (*Thiele et al., 2011*). To obtain total cell numbers, DNA of formaldehyde fixed cells filtered on 0.2 µm pore sized filters was stained with 4',6-diamidino-2-phenylindole (DAPI). Fluorescently labeled cells were subsequently counted on filter sections using an epifluorescence microscope. Likewise, bacterioplankton community composition was assessed by catalyzed reporter deposition fluorescence in situ hybridization (CARD-FISH) of formaldehyde fixed cells on

0.2 µm pore sized filters. DAPI and CARD-FISH cell counts are summarized in *Supplementary file 3* and the corresponding probes in *Supplementary file 4*.

## 16S rRNA V4 gene tag sequencing

Surface seawater samples were collected on bi-monthly to bi-weekly time scales from January 2010 to December 2012 at Helgoland roads. 500 ml of each sample were subjected to fractionating filtration as described above using 10, 3 and 0.2 µm pore size polycarbonate membrane filters (Millipore, Schwalbach, Germany). DNA of the 0.2–3 µm fraction was extracted from filters as described previously (*Sapp et al., 2007*) and quantified using the Invitrogen (Carlsbad, CA, USA) Quant-iT Pico-Green dsDNA reagent as per manufacturer's instructions. Concentrations ranged from <1 to 20 µg DNA/ml.

50 µl aliquots of each sample were pipetted into 96-well plates and sent to the Department of Energy (DOE) Joint Genome Institute (JGI, Walnut Creek, CA, USA) for amplification and sequencing as follows: Sample prep was done on a PerkinElmer (Waltham, MA, USA) Sciclone NGS G3 Liquid Handling Workstation capable of processing 96 plate-based samples in parallel, utilizing the 5 PRIME (Gaithersburg, MD 20878, USA) HotMasterMix amplification kit and custom amplification primers targeting the V4 region of the 16S rRNA gene using 515F (5' GTGCCAGCMGCCGCGGTAA 3') and 806R (5' GGACTACHVGGGTWTCTAAT 3') (*Caporaso et al., 2011*). Primers also contained Illumina adapter sequences and a barcode index. PCR reactions were set up in 75 µl with 1x HotMasterMix (5 PRIME) with final concentrations of 0.4 µg/µl BSA and 0.2 µM of each primer. This volume was split into triplicate 25 µl reactions for independent amplification and then pooled to reduce PCR bias. Prepared amplicon libraries were normalized, multiplexed into a single pool per plate and quantified using the KAPA Biosystems (Wilmington, MA, USA) next-generation sequencing library qPCR kit on a Roche (San Francisco, CA, USA) LightCycler 480. Libraries were sequenced on an Illumina (San Diego, CA, USA) MiSeq sequencer using the Reagent Kit v3 and 2x250 bp chemistry. The resulting sequences are available from the DOE-JGI GOLD database (*Reddy et al., 2015*) as part of the COGITO project (Gp0056779) and from the NCBI short read archive (SRA) (SRA278189).

## 16S rRNA gene tag analysis

Roche 454 16S rRNA gene tags from 2009 MIMAS (Microbial Interactions in Marine Systems) project (*Teeling et al., 2012*) were reanalyzed for comparison with the Illumina-based COGITO extension project from subsequent years 2010–2012. The 2009 datasets was generated using the primers Bakt_314F (5' CCTACGGGNGGCWGCAG 3') and Bakt_805R (5' GACTACHVGGGTATCTAATCC 3') (*Herlemann et al., 2011*). The forward primers of both datasets target distinct, but the reverse primers target the same region. Hence, only those 454 reads sequenced from the 805 direction were reanalyzed for comparison. For 2010–2012, raw MiSeq paired-end reads (2x250 bp) were merged and filtered using illumina-utils (https://github.com/meren/illumina-utils) to retain only read pairs without mismatches in the overlapping regions. These high-quality Illumina tags and the 454 tags were then processed separately but with the same methods via the SILVAngs pipeline (*Quast et al., 2013*), which includes additional quality filtering steps via alignment as well as length, ambiguity and homopolymer filters. Sequences were dereplicated at 100% identity and then globally clustered at 98%. Representative OTUs were classified to genus level against the SILVA (*Quast et al., 2013*) v119 database using BLAST with a similarity threshold = (sequence identity + alignment coverage) / 2 >=93%. The SAR92 clade was reclassified according to SILVA v123. Reads were mapped against representative OTUs to obtain final abundance counts. For the purpose of this study, OTUs were collapsed based on shared taxonomy no higher than the genus level.

For MIMAS samples, we retained a total of 110,995 454 reads across 7 samples with an average of 16,000 per sample. After SILVAngs quality filtering, 110,866 remained for clustering. 6102 representative OTUs were identified and 107,708 total sequences were assigned to a relative in the database during classification within the 93% similarity threshold. The final abundance matrix collapsed on shared taxonomic classification contained 500 unique taxa.

In total, 20,869,432 paired raw MiSeq reads were obtained across 142 samples from 2010–2012 COGITO samples. 15,016,350 merged reads with no mismatches in the overlapping region were retained with an average of 106,000 per sample. Reads were randomly sub-sampled to 40,000 tags per sample to reduce computational demands. In total 6,120,000 tags were submitted to the

SILVAngs pipeline. After additional quality filtering, 6,116,021 sequences were clustered at 98% and the resulting 935,006 representative OTUs were classified. A total of 5,676,259 sequences were assigned to a relative in the database within the 93% similarity threshold. The final abundance matrix collapsed on shared taxonomy no higher than the genus level contained 1995 unique taxa (*Supplementary file 5*).

## Metagenome sequencing

Total community DNA of 2009 samples (02/11/09; 03/31/09; 0407/09; 04/14/09; 06/16/09) was sequenced on the 454/Roche FLX Ti platform as described previously (*Teeling et al., 2012*). Metagenome sequencing of 2010–12 samples (03/03/2010; 04/08/10; 05/04/10; 05/18/10; 03/24/11; 04/28/11; 05/26/11; 03/08/12; 04/16/12; 05/10/12) was performed at the DOE Joint Genome Institute on the Illumina HiSeq2000 platform. Libraries were created from 100 ng environmental DNA per sample that was sheared to 270 bp using a Covaris E210 (Covaris, Woburn, MA, USA) and size selected using SPRI beads (Beckman Coulter, Indianapolis, IN, USA). The fragments were treated with end-repair, A-tailing, and ligation of Illumina compatible adapters (IDT, Coralville, IA, USA) using the KAPA-Illumina library creation kit. The libraries were quantified using KAPA Biosystem's next-generation sequencing library qPCR kit and run on a Roche LightCycler 480 real-time PCR instrument. The quantified libraries were then prepared for sequencing on the Illumina HiSeq sequencing platform utilizing a TruSeq PE Cluster Kit v3, and Illumina's cBot instrument to generate a clustered flowcell for sequencing. Sequencing was performed on the Illumina HiSeq2000 sequencer using TruSeq SBS sequencing Kits, v3, following a 2x150 bp indexed run recipe.

Raw reads were screened against Illumina artifacts with kmer size of 28, step size of 1. Reads were subsequently trimmed from both ends using a minimum quality cutoff of 3; reads with three or more N's or with average quality score <Q20 were removed. In addition, reads <50 bp were removed. The remaining quality-filtered Illumina reads were assembled using SOAPdenovo v1.05 (*Luo et al., 2012*) at a range of kmers (81, 85, 89, 93, 97, 101) with default settings (options: -K 81 -p 32 -R -d 1). Contigs generated by each assembly (6 total contig sets), were de-replicated using JGI in house Perl scripts. Contigs were then sorted into two pools based on length. Contigs <1800 bp were re-assembled using Newbler (Life Technologies, Carlsbad, CA, USA) in attempt to generate larger contigs (options: -tr, -rip, -mi 98, -ml 80). Contigs >1800 bp as well as the contigs from the Newbler assembly were combined using minimus 2 (options: -D MINID=98 -D OVERLAP=80) from the AMOS package (http://sourceforge.net/projects/amos). Read depths were estimated based on read mapping with bbmap (http://bio-bwa.sourceforge.net/). The metagenome study information is available from the DOE-JGI GOLD database (study: Gs0000079). The unassembled reads are available from the NCBI SRA (see *Supplementary file 8*), and the assembled and annotated metagenome datasets from the IMG/M system (*Markowitz et al., 2014*).

## Metagenome analysis

The DOE-JGI MAP v.4 annotation pipeline (*Huntemann et al., 2015*) was used for initial metagenome gene prediction and annotation. The annotated metagenomes were loaded in the IMG/M system as of mid 2014, and subsequently imported into a GenDB v2.2 annotation system (*Meyer et al., 2003*) for taxonomic classification and data mining.

All genes were searched against the NCBI non-redundant protein database (as of June 17th, 2014) using USEARCH v6.1.544 (*Edgar, 2010*), against the Pfam v25 database (*Finn et al., 2014*) using HMMER v3 (*Punta et al., 2012*), for signal peptides using SignalP v3.0 (*Nielsen et al., 1999*) and for transmembrane helices using TMHMM v2.0c (*Krogh et al., 2001*). CAZymes were automatically annotated based on HMMER searches against the Pfam v25 and dbCAN (*Yin et al., 2012*) databases and BLAST (*Altschul et al., 1990*) searches against the CAZy database (*Cantarel et al., 2009*; *Lombard et al., 2014*) using E-value cut-offs that were specifically adjusted for each CAZyme family (*Supplementary file 11*). Genes were only annotated as CAZymes when at least two of the search results were congruent, and CAZymes were only analyzed for contigs ≥500 bp.

Taxonomic classification of the metagenome sequences into taxonomically coherent bins ('taxobins') was carried out with a modified version of the Taxometer approach described in (*Teeling et al., 2012*). Taxometer consolidates predictions of a set of individual sequence classification tools into a consensus using a weighted assessment on seven selected ranks (superkingdom,

phylum, class, order, family, genus, species) of the NCBI taxonomy (http://www.ncbi.nlm.nih.gov/Taxonomy/). We combined taxonomic information inferred from (i) Pfam hits using the CARMA3 approach (*Gerlach and Stoye, 2011*), (ii) BLASTp hits using the KIRSTEN approach (*Teeling et al., 2012*; supp. data), and (iii) mapping of quality-filtered (illumina-utils; https://github.com/meren/illumina-utils/) Illumina reads to selected reference sequences. In contrast to the original Taxometer approach we omitted signature-based classification with Self-Organizing Maps and mapping of reads containing partial 16S rRNA gene sequences. The prediction tools that were used are outlined below:

We used the HMMER-based module of CARMA3 (not the BLAST-based module) that infers taxonomy of sequences by post-processing genes with HMMER3 hits to the Pfam database. The basic principle is to apply a reciprocal search technique to reduce the number of identified matches and thereby to improve taxonomic classification quality.

KIRSTEN (Kinship Relationship Reestablishment) infers taxonomy of sequences by post-processing BLASTp hits to the NCBI nr database by means of rank-based evaluations on all levels of the NCBI taxonomy with an increasing stringency from the superkingdom down to the species level. On each taxonomic level, all occurring taxa are weighted by the sum of their BLASTp bit scores. When the taxon with the highest weight exceeds an adjustable threshold, the process continues towards the next taxonomic level. The threshold increases with each taxonomic level, i.e. the algorithm becomes more critical while it progresses. For this study, we substituted BLASTp by the UBLAST module of USEARCH 6.1 (*Edgar, 2010*) with an E-value cutoff of E-10 and maximum hit count of 500.

SMALT (http://www.sanger.ac.uk/resources/software/smalt/) was used to map metagenome reads on a manually compiled set of 49 reference genomes and a streamlined version of the NCBI nr database. Both, the reference genomes and the sequences selected from the NCBI nr database were selected based on habitat-specific information. This was done manually for the reference genomes and automatically for the NCBI nr database as follows: Each of the hits from the UBLAST search during KIRSTEN analysis can be associated with multiple taxa, since in nr redundant sequences from different taxa are merged. We used this information to extract all taxa associated with a given hit, then combined the taxa of all hits and finally extracted all sequences of these taxa from Genbank. This way a sample-specific streamlined subset of Genbank was generated that greatly sped up the mapping process. Only metagenome reads with at least 95% identity to any sequence in the sample-specific sequence database were used. Reads were subsequently back-mapped to contigs. Since contigs consist of many reads, this way, contigs were associated with multiple taxonomic paths. Taxonomic paths resulting from classification with reference genomes and the habitat-specific streamlined nr were concatenated. Paths representing less than 1% of the reads of a given contig were discarded.

Finally, Taxometer was used to combine all gene-based predictions from CARMA3 and KIRSTEN and the mapping results for each contig, and to infer a consensus taxonomy. Taxonomic predictions were possible for 94.6% of the contigs above 1 kbp. The results are summarized in *Supplementary file 9*.

## Statistical analyses

The Spearman rank correlation test (*Supplementary file 6*) was used to test for correlations between *Bacteroidetes* clade abundances and environmental variables (chlorophyll a, temperature, salinity, silicate, phosphate, nitrate, nitrite, ammonia) and phytoplankton abundances (classes: diatoms, dinoflagellates, coccolithophorids, silicoflagellates, flagellates, ciliates, green algae; species: *Mediophyxis helysia*, *Thalassiosira nordenskioeldii*, *Chaetoceros debilis* and *C. minimus*, *Rhizosolenia styliformis*, *Chattonella* and *Phaeocystis*). *Bacteroidetes* and phytoplankton numbers were transformed to log-scale for better comparison. Linear regressions (*Supplementary file 7*) were done using stepwise forward regression model by using the log transformed *Bacteroidetes* and phytoplankton abundances. All statistical analyses were performed using the software Sigma-Plot 12 (SYSTAT, Santa Clara, CA, USA).

## Acknowledgements

The expert technical assistance of Y.-L.Chen and D. Berkelmann is acknowledged. This study was funded by the Max Planck Society. The work conducted by the US. Department of Energy Joint Genome Institute, a DOE Office of Science User Facility, is supported under Contract No. DE-AC02-05CH11231.

## Additional information

### Funding

| Funder | Author |
| --- | --- |
| Max-Planck-Gesellschaft | Rudolf I Amann |

The funders had no role in study design, data collection and interpretation, or the decision to submit the work for publication.

### Author contributions

HT, BMF, Conception and design, Acquisition of data, Analysis and interpretation of data, Drafting or revising the article; CMB, Sampling at Helgoland Island, Total cell count and CARD-FISH analyses, Acquisition of data, Analysis and interpretation of data; KK, Metagenome analyses, CAZyme analyses, Bioinformatics, Analysis and interpretation of data; MC, Analyses of the complete 16S rRNA tag dataset, Analysis and interpretation of data; LK, Metagenome analyses, Selection and analyses of reference genomes for metagenome taxonomic classification, Bioinformatics, Analysis and interpretation of data; GR, Sampling at Helgoland Island, CARD-FISH analyses, Acquisition of data, Analysis and interpretation of data; JW, Taxonomic classification of metagenomes (programming of pipeline and compilation of reference datasets), Acquisition of data, Analysis and interpretation of data; CQ, Programming of the 16S rRNA tag data analyses pipeline, Assistance with 16S rRNA tag data analyses, Analysis and interpretation of data; FOG, Conception and design, Analysis and interpretation of data; JL, Sampling at Helgoland Island, Initial analyses of a part of the 16S rRNA tag data, Acquisition of data, Analysis and interpretation of data; AW, GG, Provision of sampling logistics and lab space at Helgoland Island, Provision of physicochemical data, Provision of algae biodiversity data, Statistical analyses of 16S rRNA tag data, Acquisition of data, Analysis and interpretation of data; KHW, Provision of sampling logistics and lab space at Helgoland Island, Provision and analyses of physicochemical data, Provision and analyses of algae biodiversity data, Acquisition of data, Analysis and interpretation of data; RIA, Conception and design, Analysis and interpretation of data, Drafting or revising the article

### Author ORCIDs

Rudolf I Amann, http://orcid.org/0000-0002-0846-7372

## Additional files

### Supplementary files

• Supplementary file 1. Physicochemical parameters.

• Supplementary file 2. Cell counts of dominating phytoplankton.

• Supplementary file 3. Total cell and CARD-FISH cell counts of bacterioplankton.

• Supplementary file 4. Specific oligonucleotide probes used for quantification of free-living (0.2 - 3 μm) bacterioplankton populations by fluorescence in situ hybridization (FISH).

• Supplementary file 5. Abundance matrix of the 16S rRNA gene tag sequence analysis.

• Supplementary file 6. Spearman rank correlations of abundant *Bacteroidetes* clades to phytoplankton groups. Correlation values were considered only when p-value <0.05. Spearman rank correlations measure the strength of association between two ranked variables. A precondition (assumption) for this nonparametric statistical analysis is that the two variables share a monotonic relationship (i.e. when one variable increases, the other either consistently increases as well or consistently decreases). This monotonic relationship does, however, not need to be linear. While no particularly strong correlations were found between major algae groups and bacterioplankton clades, some noteworthy trends were detected. For example, *Ulvibacter* abundances were positively correlated with diatoms (in particular centrales) and negatively correlated with silicoflagellates, whereas an opposite trend was observed for the VIS1 clade of the NS5 marine group (i.e. negative correlation with diatoms and a positive correlation with silicoflagellates). While the explanatory power of such pairwise rank-based correlations is limited and correlation does not necessarily imply causation, these data suggest at least for *Ulvibacter* sp. (and to a lesser extent for *Formosa* clade B) that they are associated with diatoms, while the NS3a, NS5 and NS9 marine groups seemed to be rather positively correlated with flagellates such as dino- and silicoflagellates. The positive correlation between *Ulvibacter* sp. and diatoms was supported by Spearman rank correlation analysis between the abundances of prominent bacterioplankton clades and distinct phytoplankton groups. *Ulvibacter* abundances were positively correlated with the diatoms *Mediopyxis helysia*, *Chaetoceros debilis*, *Chaetoceros minimus*, *Thalassiosira nordenskioeldii* and the *Phaeocystis* spp. haptophytes. Correlations of other bacterioplankton groups with distinct phytoplankton groups were less pronounced. Noteworthy positive correlations were found between *Chattonella* spp. and *Bacteroidetes* abundances (probe CF319a), in particular *Marinoscillum* spp. (probe CYT-734), *Polaribacter* spp. (probe POL740) and the NS5/VIS1 and NS3a marine groups. The NS5/VIS1 marine group was negatively correlated with most diatom groups and the *Phaeocystis* sp. haptophytes, and only positively correlated with *Chattonella* species. This might indicate a preference for *Chattonella* spp. or simply reflect that the members of the VIS1 clade of the NS5 marine group were less competitive than other *Flavobacteriia* under the conditions where other algae dominated. Spearman rank correlations between major bacterioplankton clades and physicochemical parameters were most conclusive for *Ulvibacter* species. Ulvibacter abundances were positively correlated with chlorophyll *a*, consistent with the previously detected positive correlation with in particular diatoms. *Ulvibacter* abundances were also negatively correlated with silicate, phosphate, nitrate and ammonium. In particular the strong negative correlation with silicate, which is the limiting factor for diatom frustule formation, supports a particularly strong association of *Ulvibacter* spp. and diatoms. Similar but weaker trends were observed also for other bacterioplankton clades, such as *Polaribacter*, *Formosa* clade B, or VIS6 clade *Cryomorphaceae*. Of all genus-level clades, *Polaribacter* and *Formosa* clade A showed the strongest associations with temperature.

• Supplementary file 7. Linear regression analyses were computed in order to test, whether the abundances of major clades of *Flavobacteriia* were influenced by abiotic factors or by abundant algae groups. For all tested clades of *Flavobacteriia*, multiple abiotic factors and multiple algae groups were obtained as explanatory variables. The strongest abiotic predictors were temperature, salinity, silicate and nitrate. The strongest biotic predictors were *Phaeocystis* spp. haptophytes, *Rhizosolenia* spp., *Chaetoceros debilis*, and *Chaetoceros minimus* diatoms and the silicoflagellate *Chattonella*. It should be noted though that these regressions were computed based on log-transformed abundance data and not algae biovolumes (which were not measured). Insofar the in influence of the rather small cell-sized algae such as *Chaetoceros miniumus* is likely overestimated. Such limitations notwithstanding it is noteworthy that in no case a simple one-to-one relationship between specific algae and specific bacterioplankton groups was obtained so far.

• Supplementary file 8. Statistics of the 16 metagenomes from free-living (0.2 - 3.0 μm) North Sea bacterioplankton that were used in this study. Two additional metagenomes are listed in italics for completeness, one from a test run sampled on August 21[th], 2008, and a metagenome of particle-attached (3 - 10 μm) bacterioplankton sampled on April 14[th], 2009.

• Supplementary file 9. Results of the taxonomic classification of the metagenome sequences.

• Supplementary file 10. Sizes in basepairs of the metagenome taxonomic bins that were used for CAZyme frequency analyses on order to genus level. Taxonomic bins with too small sizes for a sound analysis were excluded (red text).

• Supplementary file 11. E-value thresholds used for automated CAZyme family detection. Searches were performed against the CAZy database, the dbCAN database and the Pfam database using E-value thresholds that were adjusted for each family by extensive manual annotations. CAZymes were only annotated when at least two of the three database searches yielded positive results (GH = glycoside hydrolase; CBM = carbohydrate- binding module; CE = carbohydrate esterase; PL = polysaccharide lyase; GT = glycoside transferase; AA = auxiliary activities).

## Major datasets

The following datasets were generated:

| Author(s) | Year | Dataset title | Dataset URL | Database, license, and accessibility information |
| --- | --- | --- | --- | --- |
| Hanno Teeling, Bernhard M Fuchs, Christin M Bennke, Karen Krüger, Meghan Chafee, Lennart Kappelmann, Greta Reintjes, Jost Waldmann, Christian Quast, Frank Oliver Glöckner, Judith Lucas, Antje Wichels, Gunnar Gerdts, Karen H Wiltshire, Rudolf I Amann | 2016 | Metagenome of North Sea surface bacteriopankton (3 - 0.2 µm) from Helgoland Roads on 2010/03/03 | http://www.ncbi.nlm.nih.gov/sra/?term= SRA212914 | Publically available at NCBI Sequence Read Archive (accession no. SRA212914) |
| Hanno Teeling, Bernhard M Fuchs, Christin M Bennke, Karen Krüger, Meghan Chafee, Lennart Kappelmann, Greta Reintjes, Jost Waldmann, Christian Quast, Frank Oliver Glöckner, Judith Lucas, Antje Wichels, Gunnar Gerdts, Karen H Wiltshire, Rudolf I Amann | 2016 | Metagenome of North Sea surface bacteriopankton (3 - 0.2 µm) from Helgoland Roads on 2010/05/04 | http://www.ncbi.nlm.nih.gov/sra/?term= SRA212889 | Publically available at NCBI Sequence Read Archive (accession no. SRA212889) |
| Hanno Teeling, Bernhard M Fuchs, Christin M Bennke, Karen Krüger, Meghan Chafee, Lennart Kappelmann, Greta Reintjes, Jost Waldmann, Christian Quast, Frank Oliver Glöckner, Judith Lucas, Antje Wichels, Gunnar Gerdts, Karen H Wiltshire, Rudolf I Amann | 2016 | 16S rRNA V4 amplicons of North Sea surface bacteriopankton (3 - 0.2 µm) from Helgoland Roads from 2010 to 2012 | http://www.ncbi.nlm.nih.gov/sra/?term= SRA278189 | Publically available at NCBI Sequence Read Archive (accession no. SRA278189) |

| | | | | |
|---|---|---|---|---|
| Hanno Teeling, Bernhard M Fuchs, Christin M Bennke, Karen Krüger, Meghan Chafee, Lennart Kappelmann, Greta Reintjes, Jost Waldmann, Christian Quast, Frank Oliver Glöckner, Judith Lucas, Antje Wichels, Gunnar Gerdts, Karen H Wiltshire, Rudolf I Amann | 2016 | Metagenome of North Sea surface bacteriopankton (3 - 0.2 μm) from Helgoland Roads on 2010/05/18 | http://www.ncbi.nlm.nih.gov/sra/?term=SRA212588 | Publically available at NCBI Sequence Read Archive (accession no. SRA212588) |
| Hanno Teeling, Bernhard M Fuchs, Christin M Bennke, Karen Krüger, Meghan Chafee, Lennart Kappelmann, Greta Reintjes, Jost Waldmann, Christian Quast, Frank Oliver Glöckner, Judith Lucas, Antje Wichels, Gunnar Gerdts, Karen H Wiltshire, Rudolf I Amann | 2016 | Metagenome of North Sea surface bacteriopankton (3 - 0.2 μm) from Helgoland Roads on 2011/03/24 | http://www.ncbi.nlm.nih.gov/sra/?term=SRA212575 | Publically available at NCBI Sequence Read Archive (accession no. SRA212575) |
| Hanno Teeling, Bernhard M Fuchs, Christin M Bennke, Karen Krüger, Meghan Chafee, Lennart Kappelmann, Greta Reintjes, Jost Waldmann, Christian Quast, Frank Oliver Glöckner, Judith Lucas, Antje Wichels, Gunnar Gerdts, Karen H Wiltshire, Rudolf I Amann | 2016 | Metagenome of North Sea surface bacteriopankton (3 - 0.2 μm) from Helgoland Roads on 2011/04/28 | http://www.ncbi.nlm.nih.gov/sra/?term=SRA212530 | Publically available at NCBI Sequence Read Archive (accession no. SRA212530) |
| Hanno Teeling, Bernhard M Fuchs, Christin M Bennke, Karen Krüger, Meghan Chafee, Lennart Kappelmann, Greta Reintjes, Jost Waldmann, Christian Quast, Frank Oliver Glöckner, Judith Lucas, Antje Wichels, Gunnar Gerdts, Karen H Wiltshire, Rudolf I Amann | 2016 | Metagenome of North Sea surface bacteriopankton (3 - 0.2 μm) from Helgoland Roads on 2011/05/26 | http://www.ncbi.nlm.nih.gov/sra/?term=SRA212476 | Publically available at NCBI Sequence Read Archive (accession no. SRA212476) |

| | | | | |
|---|---|---|---|---|
| Hanno Teeling, Bernhard M Fuchs, Christin M Bennke, Karen Krüger, Meghan Chafee, Lennart Kappelmann, Greta Reintjes, Jost Waldmann, Christian Quast, Frank Oliver Glöckner, Judith Lucas, Antje Wichels, Gunnar Gerdts, Karen H Wiltshire, Rudolf I Amann | 2016 | Metagenome of North Sea surface bacteriopankton (3 - 0.2 μm) from Helgoland Roads on 2012/03/08 | http://www.ncbi.nlm.nih.gov/sra/?term=SRA212475 | Publically available at NCBI Sequence Read Archive (accession no. SRA212475) |
| Hanno Teeling, Bernhard M Fuchs, Christin M Bennke, Karen Krüger, Meghan Chafee, Lennart Kappelmann, Greta Reintjes, Jost Waldmann, Christian Quast, Frank Oliver Glöckner, Judith Lucas, Antje Wichels, Gunnar Gerdts, Karen H Wiltshire, Rudolf I Amann | 2016 | Metagenome of North Sea surface bacteriopankton (3 - 0.2 μm) from Helgoland Roads on 2012/04/16 | http://www.ncbi.nlm.nih.gov/sra/?term=SRA212473 | Publically available at NCBI Sequence Read Archive (accession no. SRA212473) |
| Hanno Teeling, Bernhard M Fuchs, Christin M Bennke, Karen Krüger, Meghan Chafee, Lennart Kappelmann, Greta Reintjes, Jost Waldmann, Christian Quast, Frank Oliver Glöckner, Judith Lucas, Antje Wichels, Gunnar Gerdts, Karen H Wiltshire, Rudolf I Amann | 2016 | Metagenome of North Sea surface bacteriopankton (3 - 0.2 μm) from Helgoland Roads on 2012/05/10 | http://www.ncbi.nlm.nih.gov/sra/?term=SRA212472 | Publically available at NCBI Sequence Read Archive (accession no. SRA212472) |
| Hanno Teeling, Bernhard M Fuchs, Christin M Bennke, Karen Krüger, Meghan Chafee, Lennart Kappelmann, Greta Reintjes, Jost Waldmann, Christian Quast, Frank Oliver Glöckner, Judith Lucas, Antje Wichels, Gunnar Gerdts, Karen H Wiltshire, Rudolf I Amann | 2016 | Metagenome of North Sea surface bacteriopankton (3 - 0.2 μm) from Helgoland Roads on 2010/04/08 | http://www.ncbi.nlm.nih.gov/sra/?term=SRA212908 | Publically available at NCBI Sequence Read Archive (accession no. SRA212908) |

The following previously published datasets were used:

| Author(s) | Year | Dataset title | Dataset URL | Database, license, and accessibility information |
|---|---|---|---|---|
| Hanno Teeling, Bernhard M. Fuchs, Dörte Becher, Christine Klockow, Antje Gardebrecht, Christin M. Bennke, Mariette Kassabgy, Sixing Huang, Alexander J. Mann, Jost Waldmann, Marc Weber, Anna Klindworth, Andreas Otto, Jana Lange, Jörg Bernhardt, Christine Reinsch, Michael Hecker, Jörg Peplies, Frank D. Bockelmann, Ulrich Callies, Gunnar Gerdts, Antje Wichels, Karen H. Wiltshire, Frank Oliver Glöckner, Thomas Schweder, Rudolf Amann | 2012 | Metagenome of North Sea surface bacteriopankton (3 - 0.2 µm) from Helgoland Roads on 2009/02/11 | http://www.ebi.ac.uk/ena/data/view/PRJEB2888 | Publically available at the European Nucleotide Archive (study accession no. PRJEB2888, experimental accession no. ERX069460 - ERX069464) |
| Hanno Teeling, Bernhard M. Fuchs, Dörte Becher, Christine Klockow, Antje Gardebrecht, Christin M. Bennke, Mariette Kassabgy, Sixing Huang, Alexander J. Mann, Jost Waldmann, Marc Weber, Anna Klindworth, Andreas Otto, Jana Lange, Jörg Bernhardt, Christine Reinsch, Michael Hecker, Jörg Peplies, Frank D. Bockelmann, Ulrich Callies, Gunnar Gerdts, Antje Wichels, Karen H. Wiltshire, Frank Oliver Glöckner, Thomas Schweder, Rudolf Amann | 2012 | Metagenome of North Sea surface bacteriopankton (3 - 0.2 µm) from Helgoland Roads on 2009/09/01 | http://www.ebi.ac.uk/ena/data/view/PRJEB2888 | Publically available at the European Nucleotide Archive (study accession no. PRJEB2888, experimental accession no. ERX069492 - ERX069495) |

| | | | | |
|---|---|---|---|---|
| Hanno Teeling, Bernhard M. Fuchs, Dörte Becher, Christine Klockow, Antje Gardebrecht, Christin M. Bennke, Mariette Kassabgy, Sixing Huang, Alexander J. Mann, Jost Waldmann, Marc Weber, Anna Klindworth, Andreas Otto, Jana Lange, Jörg Bernhardt, Christine Reinsch, Michael Hecker, Jörg Peplies, Frank D. Bockelmann, Ulrich Callies, Gunnar Gerdts, Antje Wichels, Karen H. Wiltshire, Frank Oliver Glöckner, Thomas Schweder, Rudolf Amann | 2012 | Metagenome of North Sea surface bacteriopankton (3 - 0.2 µm) from Helgoland Roads on 2009/06/16 | http://www.ebi.ac.uk/ena/data/view/PRJEB2888 | Publically available at the European Nucleotide Archive (study accession no. PRJEB2888, experimental accession no. ERX069490 - ERX069491) |
| Hanno Teeling, Bernhard M. Fuchs, Dörte Becher, Christine Klockow, Antje Gardebrecht, Christin M. Bennke, Mariette Kassabgy, Sixing Huang, Alexander J. Mann, Jost Waldmann, Marc Weber, Anna Klindworth, Andreas Otto, Jana Lange, Jörg Bernhardt, Christine Reinsch, Michael Hecker, Jörg Peplies, Frank D. Bockelmann, Ulrich Callies, Gunnar Gerdts, Antje Wichels, Karen H. Wiltshire, Frank Oliver Glöckner, Thomas Schweder, Rudolf Amann | 2012 | Metagenome of North Sea surface bacteriopankton (3 - 0.2 µm) from Helgoland Roads on 2009/04/14 | http://www.ebi.ac.uk/ena/data/view/PRJEB2888 | Publically available at the European Nucleotide Archive (study accession no. PRJEB2888, experimental accession no. ERX069474 - ERX069481) |

| | | | | |
|---|---|---|---|---|
| Hanno Teeling, Bernhard M. Fuchs, Dörte Becher, Christine Klockow, Antje Gardebrecht, Christin M. Bennke, Mariette Kassabgy, Sixing Huang, Alexander J. Mann, Jost Waldmann, Marc Weber, Anna Klindworth, Andreas Otto, Jana Lange, Jörg Bernhardt, Christine Reinsch, Michael Hecker, Jörg Peplies, Frank D. Bockelmann, Ulrich Callies, Gunnar Gerdts, Antje Wichels, Karen H. Wiltshire, Frank Oliver Glöckner, Thomas Schweder, Rudolf Amann | 2012 | Metagenome of North Sea surface bacteriopankton (3 - 0.2 μm) from Helgoland Roads on 2009/04/07 | http://www.ebi.ac.uk/ena/data/view/PRJEB2888 | Publically available at the European Nucleotide Archive (study accession no. PRJEB2888, experimental accession no. ERX069470 - ERX069473) |
| Hanno Teeling, Bernhard M. Fuchs, Dörte Becher, Christine Klockow, Antje Gardebrecht, Christin M. Bennke, Mariette Kassabgy, Sixing Huang, Alexander J. Mann, Jost Waldmann, Marc Weber, Anna Klindworth, Andreas Otto, Jana Lange, Jörg Bernhardt, Christine Reinsch, Michael Hecker, Jörg Peplies, Frank D. Bockelmann, Ulrich Callies, Gunnar Gerdts, Antje Wichels, Karen H. Wiltshire, Frank Oliver Glöckner, Thomas Schweder, Rudolf Amann | 2012 | Metagenome of North Sea surface bacteriopankton (3 - 0.2 μm) from Helgoland Roads on 2009/03/31 | http://www.ebi.ac.uk/ena/data/view/PRJEB2888 | ERX069465 - ERX069469 |

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
