## [Decision Letter]

[Editors’ note: this article was originally rejected after discussions between the reviewers, but the authors were invited to resubmit after an appeal against the decision.]

Thank you for submitting your work entitled "Recurring patterns in bacterioplankton dynamics during coastal spring algae blooms" for consideration by *eLife*. Your article has been favorably evaluated by Ian Baldwin (Senior editor) and three reviewers, one of whom is a member of our Board of Reviewing Editors. Our decision has been reached after consultation between the reviewers. Based on these discussions and the individual reviews below, we regret to inform you that your work in its current format will not be considered further for publication in *eLife*.

We think you have submitted a very interesting study and the reviewers were collectively enthusiastic about the efforts to collect the data. However, at the same time the reviewers also were concerned about the novelty yet provided compared to the Science paper of 2012. They felt that a more rigorous analysis will yield new and interesting insights into bacterial phytoplankton interactions. Furthermore, the reviewers indicated some experiments and also experimental and methodological improvements, which might lead to a study with deeper insight into the interaction of algae and bacteria, but this cannot be in the time schedule we expect for a major revision.

*Reviewer #1:*

The manuscript describes the characterisation of planktonic blooms following the primary blooms of algae in the southern North Sea. Such successions during spring phytoplankton blooms were investigated for four consecutive years. An enormous amount of data was generated by dense sampling and high-resolution taxonomic analyses. These data are of very high quality. In addition, metagenomic data were provided which revealed recurring patterns at the functional level, in particular concerning polysaccharide degradation. The manuscript is very well written, the outcome is very interesting. The methods used are state of the art. I have two comments to be discussed:

1) The functional analysis is purely based on occurrence of e.g. CAZyme-encoding genes by metagenome sequencing. Of course, the findings are sound, i.e., a succession of bacteria with a distinct set of CAZyme-encoding genes was detected. I am not sure whether it would be possible to also carry out a proteome analysis to demonstrate the presence of these enzymes, many of them can be expected to be extracellular. I am not sure whether this is possible at all.

2) I think the manuscript contains sufficient novelty compared to the Science paper of 2012, where some of the authors showed that substrate-controlled succession of marine bacterioplankton populations is induced by a phytoplankton bloom. In the current manuscript a far more comprehensive analysis was provided.

In general, I think the manuscript reports very interesting data, and also functional relationships. Also, I expect the data being an extremely valuable source for further studies.

*Reviewer #2:*

This is a significant contribution to the understanding of how bacteria respond to spring phytoplankton blooms. It appears technically sound in design and data reporting, and provides great detail that will inform our understanding of this important phenomenon. My main concern: There are clearly consistencies in the bacterial responses yearly, however the primary conclusion that there are "recurring patterns" of bacterioplankton dynamics is not statistically supported, and frankly vague. A clear quantitative (and justified) definition is needed, otherwise one can focus on what parts repeat and say there is a repeating pattern, or focus on differences and say there isn't. The succession details in their earlier Science paper (Teeling et al., 2012) differ here. If one accepts a broad qualitative definition of "recurring patterns," the results are confirmatory of previous work by the authors and others, cited by recent reviews (Buchan et al., Nature Reviews Microbiology, for example), showing that many taxa, including those reported here, have repeated seasonality or bloom responses. Admittedly few prior studies covered the fine temporal scale measured here, and that is a considerable strength.

To quantitatively show dynamics are recurring, statistics should be applied using each year as a sort of replicate to determine which aspects of the microbial successions repeat, including relationships to algal taxa. The spring "blooms" are quite variable in their dynamics, as acknowledged by the authors, and the response differs, as expected. If bottom up controls are the dominant factor controlling the bacterial composition as suggested, the why are dynamics not similar within a year when there is a secondary peak of chlorophyll after the first bloom (see specific comments)? Could it be top-down controls?

Regarding associations (or lack thereof) with phytoplankton, I think the assessment of the phytoplankton dynamics is not universal enough to allow full conclusions. For instance, only one member an important group of phytoplankton, the haptophytes, is reported (*Phaeocystis*) – are there no others? Additionally, among the most numerically abundant taxa were the Green Algae, which have many significantly different species, but here they are "lumped" together. It is unlikely that particular microbial interactions would occur with whole families or orders of phytoplankton, but instead be rather very specific (e.g. see Amin et al. 2015 and references therein). Unfortunately, the manuscript only reports a relatively small number of phytoplankton species – and it would be good to see calculations of the fraction of total phytoplankton these represent (e.g. biomass totals to compare with chlorophyll-based totals to see what% may be missing. Some chlorophyll increases were not accompanied on a daily basis with counts). Furthermore, the main way the authors have looked for correlation between phytoplankton and bacteria is via Spearman's correlation for apparently all of the years together. By visual examination, many of the "responding" bacteria seem to have a delayed reaction to the increases in chlorophyll, so the authors should also look for time-delayed correlations, in addition to year-by-year studies.

CARD-FISH identified only about 70% of the bacteria, a shortcoming to understanding the full community response, addressable with their less frequent 16S observations. The broad groups appear to have little consistent response to the 'blooms.' Why no direct comparisons between CARD-FISH and 16S sequences? Some groups explored with the 16S not found in CARD-FISH are generally ignored, but CARD-FISH could represent as little as about 40% during blooms. Notable such bacteria include *Verrucomicrobia* (increases in May 2010, 2011,2012). Also, chloroplast data should be separated from figures of 16S tag sequencing results because they are not bacteria or archaea.

Important: Is the variability observed largely temporal (local successions) or is it a mixture of spatial and temporal variability (most likely)? Are the blooms very patchy or regional? Satellite imagery would help, even occasional. Minimally – deserves discussion.

*Reviewer #3:*

This manuscript describes an impressive campaign to characterize the bacterial communities in a coastally influenced environment over the course of consecutive several years. These types of long-term, high-resolution temporal studies have been previously restricted to more oligotrophic environments. Here, the authors focus their analysis on bacterial communities associated with the seasonal phytoplankton blooms that occur in their study site, with the broader goal of providing a better understanding of the underlying drivers of the recurrence of specific bacterial lineages/groups.

The scientific methods are solid, and I have no major worries regarding the quality of the data. My main concern (indeed, personal struggle) is that I am unsure of the scientific impact of this work, at least as currently presented in the manuscript. While this is a truly impressive dataset and one that should be published/disseminated, the main conclusions are that some bacterial lineages (and some of the functions they encode) show recurring seasonal and annual patterns of occurrence. Linkages between specific bacterial taxa, functions and environmental parameters (principally those associated with phytoplankton blooms) are evident. Yet, as the authors report, this has been shown previously; albeit these "elsewhere" environments are generally more oligotrophic, and perhaps subject to less environmental variability (though that is debatable) than the one characterized here. The authors frame their conclusions on the nature of the succession of bacterial groups as being one that is influenced by both "stochastic processes" and "deterministic principles", however, I did not find the manuscript to be convincingly clear on what they mean by these statements.

Given the criteria for acceptance in this journal – "highly influential research" –, I am of the position that the authors need to dig deeper to draw more novel conclusions with respects to what this time-series analysis reveals regarding environmental parameters and bacterial structure and function. The volume and nature of these data suggest that would be appropriate for some modeling efforts. This type of exercise might better support the authors conclusions and illustrate the emergent properties/recurrent patterns of specific bacterial lineages.

[Editors’ note: what now follows is the decision letter after the authors submitted for further consideration.]

Thank you for resubmitting your work entitled "Recurring patterns in bacterioplankton dynamics during coastal spring algae blooms" for further consideration at *eLife*. Your revised article has been favorably evaluated by Ian Baldwin (Senior editor) and three reviewers, one of whom is a member of our Board of Reviewing Editors. The manuscript has been improved but there are some remaining issues that need to be addressed before acceptance, as outlined below:

Although we acknowledge that the revised version was considerably improved we still felt that some points made by the reviewers were valid and simply disagreed by the authors. On the other side, we also see that the dataset is very valuable and that the authors had made more new, convincing, and quantitative conclusions. There are several points which need to be addressed. In addition to those outlined by the reviewers. We ask you to tone down the language associated with the conclusions, especially those related to the presumptive underlying drivers of the observed patterns ("substrate-induced forcing"). This conclusion is based on metagenomic analysis and functions that have not been shown conclusively. We would like to ask you to modify the text accordingly. For instance, in the Abstract "We thereby demonstrate that even though there is substantial inter-annual variation between spring phytoplankton blooms, the accompanying succession of bacterial clades is governed by deterministic principles such as substrate-induced forcing" could be modified to something like: "We thereby hypothesize that even though…".

*Reviewer #1:*

The study reports the characterisation of planktonic blooms following the primary blooms of algae in the southern North Sea. Successions during spring phytoplankton blooms have been studied for four consecutive years. An enormous amount of data was collected by dense sampling and high-resolution taxonomic analyses. As I wrote in my first report, these data are of very high quality. In addition, metagenomic data were provided which revealed recurring patterns at the functional level, in particular concerning polysaccharide degradation.

The revised version of the manuscript was considerably improved. At least to me, many questions were well addressed. The manuscript is very well written, the outcome is very interesting. The methods used are state of the art. The discrepancy of data obtained by CARD-FISH and 16S rRNA tag sequencing was well discussed.

The authors added additional data such as the full set of physicochemical measurements and the data of the dominating algae taxa. The number of figures is extraordinary, but I agree with the authors, that is required to show the full data set in a reasonable manner. This is one of the reasons, why I expect the data being an extremely valuable source for further studies.

*Reviewer #2:*

We thank the authors for their response to our criticism. We appreciate their detailed and extended response to many of our technical suggestions, of which they responded to compellingly. The dataset and extensive analysis is indeed very impressive and valuable. However, we were surprised and disappointed that the authors made relatively few changes in their manuscript in response to our main concerns, so our view of the manuscript in terms of the novelty of general conclusions and quantitative assessment of "recurring patterns" has not significantly changed. We regret we cannot be more positive after reading their response.

We do now better understand what the authors mean by "recurring patterns,", i.e., that many clades appear at >5% abundance more than once following a spring bloom. However, we find this general observation itself to not be exceptionally novel. Note, for example, in the abstract as background for the study, the authors state that *Flavobacteria, Gammaproteobacteria*, and alphaproteobacteria are known associated with blooms (Abstract). Earlier published studies include reports where the same taxa are found in different years, including after blooms (admittedly with much less comprehensive overall system analysis than this study).

Specific suggestion: In the figures they should use arrows or other notation to point out the recurring clades and when they bloom.

*Reviewer #3:*

I feel that the revised manuscript is significant improved with respects to readability and articulation of the importance of the work.

---

## [Author Response]

[Editors’ note: the author responses to the first round of peer review follow.]

*We think you have submitted a very interesting study and the reviewers were collectively enthusiastic about the efforts to collect the data. However, at the same time the reviewers also were concerned about the novelty yet provided compared to the Science paper of 2012. They felt that a more rigorous analysis will yield new and interesting insights into bacterial phytoplankton interactions. Furthermore, the reviewers indicated some experiments and also experimental and methodological improvements, which might lead to a study with deeper insight into the interaction of algae and bacteria, but this cannot be in the time schedule we expect for a major revision.* Reviewer #1:

*The manuscript describes the characterisation of planktonic blooms following the primary blooms of algae in the southern North Sea. Such successions during spring phytoplankton blooms were investigated for four consecutive years. An enormous amount of data was generated by dense sampling and high-resolution taxonomic analyses. These data are of very high quality. In addition, metagenomic data were provided which revealed recurring patterns at the functional level, in particular concerning polysaccharide degradation. The manuscript is very well written, the outcome is very interesting. The methods used are state of the art. I have two comments to be discussed:*

*1) The functional analysis is purely based on occurrence of e.g. CAZyme-encoding genes by metagenome sequencing. Of course, the findings are sound, i.e., a succession of bacteria with a distinct set of CAZyme-encoding genes was detected. I am not sure whether it would be possible to also carry out a proteome analysis to demonstrate the presence of these enzymes, many of them can be expected to be extracellular. I am not sure whether this is possible at all.*

We did carry out ten metaproteome analyses of soluble fractions of bacterioplankton cells that correspond to the 10 new metagenomes of 2010 to 2012 that we present in our study. The smallest metaproteome dataset of the 2010 to 2012 blooms comprises only 1,058 and the largest 3,994 proteins (25,832 proteins for all 10 time points combined). Since the blooms of 2010 to 2012 were also less intense than the bloom of 2009, these metaproteomes yielded mostly conventional highly expressed proteins such as chaperonins, ribosomal proteins and transporters. While the proteomes were not deep enough to detect many expressed CAZymes, we could however, as in 2009, detect remarkably high levels of TonB-dependent outer membrane receptors/transporters among the proteins with the highest expression levels. We will use this expression of TonB-dependent receptors as an indirect measure for the expression of CAZyme-encoding polysaccharide utilization loci (PULs) in a future study. In order to achieve this, we have systematically isolated and sequenced close to 50 genomes of North Sea *Flavobacteriia*. We are working on a systematic investigation of the co-localization of TonB-dependent receptors (TBDRs) and CAZymes in PULs of these isolates and predict polysaccharide substrate specificities. We will subsequently use the metaproteome data in conjunction with these (meta-)genomic data to use TBDR expression as a proxy for expression of CAZymes. This is a much more laborious, but also targeted approach, which has the potential for additional insights on relevant polysaccharide substrates, which we plan to publish in a separate paper.

*2) I think the manuscript contains sufficient novelty compared to the Science paper of 2012, where some of the authors showed that substrate-controlled succession of marine bacterioplankton populations is induced by a phytoplankton bloom. In the current manuscript a far more comprehensive analysis was provided. In general, I think the manuscript reports very interesting data, and also functional relationships. Also, I expect the data being an extremely valuable source for further studies.* We would like to thank reviewer #1 for this statement. The dataset we present is indeed unusually large (e.g. 10x more metagenomic information than in the 2012 Science paper), which prevents analysis of every possible aspect. Instead, it is our intention to explain the dataset and to focus the analyses of biodiversity and gene function repertoires on selected relevant key aspects in continuation of our initial 2012 Science publication.

We agree with reviewer #1 that the dataset will allow us and others to explore additional aspects, for example the diversity and genomic potential of bacterioplankton groups that were not in our focus, such as for example members of the alphaproteobacterial *Roseobacter* or the betaproteobacterial OM43 clades, or to focus on other relevant gene functions besides those involved in the concerted decomposition of polysaccharides.

In order to make the dataset more accessible, we now include the full set of physicochemical measurements and the data of the dominating algae taxa as supplementary material (new [Supplementary-material SD1-data] and [Supplementary-material SD2-data]).

Reviewer #2:

*This is a significant contribution to the understanding of how bacteria respond to spring phytoplankton blooms. It appears technically sound in design and data reporting, and provides great detail that will inform our understanding of this important phenomenon. My main concern: There are clearly consistencies in the bacterial responses yearly, however the primary conclusion that there are "recurring patterns" of bacterioplankton dynamics is not statistically supported, and frankly vague. A clear quantitative (and justified) definition is needed, otherwise one can focus on what parts repeat and say there is a repeating pattern, or focus on differences and say there isn't.*

We are communicating a very densely sampled large dataset and for reasons of practicality we focused on the major clades. Within these clades, there is a clear pattern of recurrence, not in the sense that the order of clades is identical each year (which is not expected due to variations in phytoplankton dynamics), but in the sense that these clades recur from year to year, which we prefer to show as original data.

We used a quantitative criterion of equal or larger than 5% relative abundance in at least two of the analyzed four for the most abundant clades, and some of the clades (defined mostly on the level of genus) that we describe responded with even much higher peaks. We made this more clear in the revised manuscript (subsection “bacterioplankton - diversity and bloom characteristics”, fourth paragraph). Annual recurrence of these clades is independent of any stochastic occurrence of other clades, and this means that we did not hand-pick results in order to make up recurrence as this reviewer implies. We focused our study further to clades from *Bacteroidete*s and *Gammaproteobacteria*, two clades known for their potential to degradepolysaccharides and demonstrated in the 2012 Teeling paper to respond to phytoplankton blooms. We only touch upon other abundant groups such as SAR11, *Roseobacter* or OM43 that lack a specific potential for polysaccharide degradation.

*The succession details in their earlier Science paper (Teeling et al., 2012) differ here. If one accepts a broad qualitative definition of "recurring patterns," the results are confirmatory of previous work by the authors and others, cited by recent reviews (Buchan et al., Nature Reviews Microbiology, for example), showing that many taxa, including those reported here, have repeated seasonality or bloom responses. Admittedly few prior studies covered the fine temporal scale measured here, and that is a considerable strength.*

The purpose of this study was to test our hypothesis that spring phytoplankton blooms trigger rapid successions of flavobacterial and gammaproteobacterial clades specialized for polysaccharide degradation and to investigate this succession in a very comprehensive dataset collected in four consecutive years.

To the best of our knowledge so far nobody has performed a polyphasic study of this scope including FISH cell counts, diversity assessment by tag sequencing, and metagenomic profiling for the same study site over a multiple year time span. To cite the 2014 review of Alison Buchan et al. that this reviewer brings forward:

"However, such studies often differ with respect to bloom location, duration and intensity of sampling, phytoplankton species succession, geochemistry and temperature, which prevents quantitative generalizations of bacterial composition and responses."

Here we address exactly this problem by presenting a high temporal resolution study from one and the same location that spans four consecutive years.

*To quantitatively show dynamics are recurring, statistics should be applied using each year as a sort of replicate to determine which aspects of the microbial successions repeat, including relationships to algal taxa.*

As stated above, the recurrence of abundant taxa can visually be seen in the data we show. We actually selected *eLife* because it is an electronic journal able to accommodate significantly more of our high-resolution original data than a printed journal. Any statistical approach with n=4, i.e. four years as replicates, has severe limitations, yet we show recurrence of specific clades in 3-4 out of 4 years. Likewise, we could further extend the analysis that we already did with respect to the relationships of specific bacterial genera to algal taxa. However, we would like to make this reviewer aware of the fact that our study deals with free-living bacteria in the size fraction 0.2 – 3 µm and not with bacteria that are directly algae-associated such as phycosphere bacteria. In any case, the statistical analyses provided show that there is no simple one to one relationship between occurrence of dedicated algae taxa and dedicated clades of free-living bacteria. This is actually a major new result of this study, consistent with our hypothesis that the availability of substrates commonly occurring in microalgae such as laminarin are a major cause for the bacterial succession, and more important than one-to-one interactions of particular phytoplankton and bacterioplankton species (see Discussion, eighth paragraph). This is now also stated in the final sentence of the concluding remarks (see end of Discussion).

*The spring "blooms" are quite variable in their dynamics, as acknowledged by the authors, and the response differs, as expected. If bottom up controls are the dominant factor controlling the bacterial composition as suggested, the why are dynamics not similar within a year when there is a secondary peak of chlorophyll after the first bloom (see specific comments)? Could it be top-down controls?* We designed the study in a way that it is strictly focused on the first spring bloom phase, since at the end of the winter period top-down factors should beminimal. Hence, we rephrased as follows:

“Dominance of bottom-up over top-down control is assumed to be characteristic for the initial phases of spring phytoplankton blooms. […] Hence, top-down control by predation sets in only during later bloom phases. This situation is distinct from summer and fall phytoplankton blooms.”

By no means we stated that bottom-up control is the dominant factor for all taxa for the entire year, but bottom-up control likely dominates during early bloom stages when predators and viruses are not yet abundant. Of course top-down control by grazing or virus mortality cannot be excluded, in particular during later bloom phases in which successions would be much more difficult to predict.

*Regarding associations (or lack thereof) with phytoplankton, I think the assessment of the phytoplankton dynamics is not universal enough to allow full conclusions. For instance, only one member an important group of phytoplankton, the haptophytes, is reported (Phaeocystis) – are there no others?*

In fact, we have very detailed phytoplankton data for all four years, and besides *Phaeocystis*, we discuss *Chaetoceros debilis, Chaetoceros minimus, Mediopyxis helysia, Rhizosolenia* styliformis and *Thalassiosira nordenskioeldii, Chattonella spp.,* and *Phaeocystis* spp. as dominant groups.Likewise, all of these groups and the major phytoplankton clades were statistically tested for association. We added the respective primary data as new [Supplementary-material SD2-data].

*Additionally, among the most numerically abundant taxa were the Green Algae, which have many significantly different species, but here they are "lumped" together.*

Green algae simply did not play a major role during the North Sea spring phytoplankton blooms that we analyzed, which in most cases are dominated by diatoms. This is why we did not include any information on this paraphyletic group in the abundance plots (Figure 3) and treated them as one group in statistical analyses. We included the primary algae count data (new [Supplementary-material SD2-data]) for reference that shows that the numbers of green algae were minute compared to the major algae taxa. We do have high-resolution data of many more algae taxa at hand (more than 200 algae taxa are continuously identified at Helgoland in the framework of the Helgoland Roads time series since 1962), and these data are available on request. (see: https://www.awi.de/forschung/biowissenschaft/oekologie-der-kuesten/arbeitsgruppen/ag-langzeitoekologieforschung-old-version/long-term-observations/helgoland-roads.html)

*It is unlikely that particular microbial interactions would occur with whole families or orders of phytoplankton, but instead be rather very specific (e.g. see Amin et al. 2015 and references therein). Unfortunately, the manuscript only reports a relatively small number of phytoplankton species – and it would be good to see calculations of the fraction of total phytoplankton these represent (e.g. biomass totals to compare with chlorophyll-based totals to see what% may be missing. Some chlorophyll increases were not accompanied on a daily basis with counts).*

We concur that associations of algae and phycosphere bacteria can be quite specific, but those bacteria were not subject of our study in which we examined the 0.2 – 3 µm fraction. We did include species level assignments of phytoplankton that was dominant in terms of biomass (Figure 3 bold lines) except for the raphidophyte *Chattonella* that dominated in 2012. In this context we would like to point out to this reviewer that the 2012 bloom was distinct from the usually diatom-dominated spring blooms at Helgoland, and still we found the same bacterioplankton clades responding. This is a clear indication against the proposed strong effect of algal identity on the sampled 0.2 – 3 µm bacterioplankton fraction. It is actually among the main outcomes of this study that indeed families and orders of phytoplankton and not particular species are important during spring blooms in the Southern North Sea, i.e. all diatoms and several other microalgae have laminarin as the main storage polysaccharide.

We are sorry that within this multi-year study we were not able to provide daily measurements for all parameters. As we explained in the manuscript, there is no strict one-to-one correlation between algae biomass and chlorophyll *a*. Reliable biomass data is very hard to obtain, and this reviewer certainly agrees that sizes of algae depend heavily on the physiological state. There are of course published values for all of the major taxa that occurred during the blooms we investigated, but usage of such values would be very rough estimations with a considerable margin of error, which is why we did abstain from including such error-prone biomass data in statistical analyses.

*Furthermore, the main way the authors have looked for correlation between phytoplankton and bacteria is via Spearman's correlation for apparently all of the years together. By visual examination, many of the "responding" bacteria seem to have a delayed reaction to the increases in chlorophyll, so the authors should also look for time-delayed correlations, in addition to year-by-year studies.*

We agree that a delayed reaction of the responding bacteria is obvious. For a brief analysis of such delays, lagged correlation approaches (LSA, localsimilarity analysis) could be applied generally. However, for these analyses, consistent and coherent datasets are required without any missing values. Due to inherent limitations of field sampling, such datasets are scarce and only few studies fulfilling the LSA requirements were published until now. In the context of our study consistency was hard to achieve due to the unpredictability of the exact onset of the spring phytoplankton blooms.

We did in fact compute forward regressions for each year that yielded no significant correlations. Only a lagged co-occurrence could be proven since the interrelation-mediating metabolites (e.g. polysaccharides) were not analyzed, since this is technically extremely demanding. Future studies will need to include quantitative analyses of structurally resolved polysaccharides.

*CARD-FISH identified only about 70% of the bacteria, a shortcoming to understanding the full community response, addressable with their less frequent 16S observations. The broad groups appear to have little consistent response to the 'blooms.' Why no direct comparisons between CARD-FISH and 16S sequences?*

CARD-FISH does have the decided advantage over 16S rRNA tag sequencing that it provides reliable cell counts. The disadvantage is that it is a laborious technique when done for a very large number of samples, which is why regrettably few studies include such data. In contrast to many other high-profile microbiome studies we actually provide exact numbers for those clades expected to recur. In order to counteract the disadvantages that we only count what we are probing for by CARD-FISH, we have investigated all samples by in-depth 16S rRNA tag sequencing to also identify novel clades. The tag data that we provide represent a dense series for 2010 to 2012 and all of these samples were deeply sequenced (rarefaction analyses not included in the manuscript show that we sequenced multiple times beyond exhaustion of any increase in novel taxa).

As for a direct comparison of CARD-FISH and tag data, we would like to remind this reviewer that perfect consensus of both methods is not possible for fundamental reasons, because the taxon frequency obtained by tag sequencing are biased among other factors by differential PCR amplification. Also CARD-FISH probe specificities and taxonomic affiliations based on tag sequencing do not match perfectly. This is well known and inherent to the way both technologies work. We advise to use the probe match tool implemented in the Silva database for an exact evaluation of the groups that the CARD-FISH probes that we used target. All of these probes are published (see the references that we provided), including information on their specificities. A technical discussion of multiple reasons why CARD-FISH numbers and tag frequencies might differ is beyond the scope of this manuscript.

*Some groups explored with the 16S not found in CARD-FISH are generally ignored, but CARD-FISH could represent as little as about 40% during blooms. Notable such bacteria include Verrucomicrobia (increases in May 2010, 2011,2012). Also, chloroplast data should be separated from figures of 16S tag sequencing results because they are not bacteria or archaea.* This is a hypothesis-driven study in which we indeed monitored particular bacterial clades using of CARD-FISH. The reviewer is thus correct that we did not include all abundant bacterial groups in the Discussion. Since the intention of this paper was not the general assessment of all aspects of bacterial diversity we in particular “neglected” groups such as SAR11, OM43 or *Roseobacter*, which are known to lack significant polysaccharide degradationpotential. A complete discussion of all data retrieved is simply impossible for a dataset of such size and hence we focused on key aspects that we think are prominent and of major importance. However, based on the reviewer’s remark we included a short comment on *Verrucomicrobia* (subsection “bacterioplankton - diversity and bloom characteristics”, third paragraph), which are for sure a very interesting group that responded to the investigated spring phytoplankton blooms with decreasing maximum abundances of 7.7% (2010), 5.0% (2011) and 2.9% (2012). This decrease correlates with decreasing bloom intensities, which agrees with a proposed role of *Verrucomicrobia* in polysaccharide decomposition (e.g. Manuel Martinez-Garcia et al., 2012).

As suggested we removed the chloroplast data and all tags ascribed as *Eukaryota* from the calculations of tag frequencies and recalculated all graphsin the respective figure (now Figure 4) accordingly.

*Important: Is the variability observed largely temporal (local successions) or is it a mixture of spatial and temporal variability (most likely)? Are the blooms very patchy or regional? Satellite imagery would help, even occasional. Minimally – deserves discussion.*

Based on the reviewer’s remarks we added satellite data (new Figure 2), which we already had analyzed, yet did not include. These data indicate that the variability is largely temporal, but there are of course also spatial influences. We discussed this in detail for the Helgoland sampling site in our Science publication of the 2009 spring bloom using satellite images and particle drift simulations. Particle drift simulation data could be computed and made available on request, but would further inflate the dataset.

[Editors’ note: the author responses to the re-review follow.]

The manuscript has been improved but there are some remaining issues that need to be addressed before acceptance, as outlined below: Although we acknowledge that the revised version was considerably improved we still felt that some points made by the reviewers were valid and simply disagreed by the authors. On the other side, we also see that the dataset is very valuable and that the authors had made more new, convincing, and quantitative conclusions. There are several points which need to be addressed. In addition to those outlined by the reviewers. We ask you to tone down the language associated with the conclusions, especially those related to the presumptive underlying drivers of the observed patterns ("substrate-induced forcing"). This conclusion is based on metagenomic analysis and functions that have not been shown conclusively. We would like to ask you to modify the text accordingly. For instance, in the Abstract "We thereby demonstrate that even though there is substantial inter-annual variation between spring phytoplankton blooms, the accompanying succession of bacterial clades is governed by deterministic principles such as substrate-induced forcing" could be modified to something like: "We thereby hypothesize that even though…".

We agree that our data does not provide a direct proof of deterministic effects such as substrate-induced forcing, but rather provides indirect indications that such effects might play a major role during spring bloom successions. We hence toned down the language of our conclusions throughout the manuscript as requested. The above-mentioned part of the Abstract was rephrased to:

"We therefore hypothesize that even though there is substantial inter-annual variation between spring phytoplankton blooms, the accompanying succession of bacterial clades is largely governed by deterministic principles such as substrate-induced forcing."

In the Results section, we rephrased as follows:

"Metagenome taxonomic classification provided sufficient data for analysis of CAZyme repertoires of the flavobacterial NS5 marine group, NS3a marine group, *Formosa, Polaribacter*, and *Cryomorphaceae*, and the gammaproteobacterial *Alteromonadales* and *Reinekea* clades ([Supplementary-material SD10-data]). For most of these clades, the analyses revealed fingerprint-like patterns, which corroborates the hypothesis that these clades have distinct glycan niches that are relatively stable across years.*"*

In the Discussion, we rephrased as follows:

"The existence of recurrent key players during North Sea spring phytoplankton blooms suggests that the bacterioplankton community composition during and after such blooms is governed by deterministic effects."

And finally in the concluding remarks, we rephrased the respective statement as follows:

"Our results suggest that besides stochastic also deterministic effects influence phytoplankton-bacterioplankton coupling during blooms."

Reviewer #2:

*We thank the authors for their response to our criticism. We appreciate their detailed and extended response to many of our technical suggestions, of which they responded to compellingly. The dataset and extensive analysis is indeed very impressive and valuable. However, we were surprised and disappointed that the authors made relatively few changes in their manuscript in response to our main concerns, so our view of the manuscript in terms of the novelty of general conclusions and quantitative assessment of "recurring patterns" has not significantly changed. We regret we cannot be more positive after reading their response. We do now better understand what the authors mean by "recurring patterns,", i.e., that many clades appear at >5% abundance more than once following a spring bloom. However, we find this general observation itself to not be exceptionally novel. Note, for example, in the abstract as background for the study, the authors state that Flavobacteria, Gammaproteobacteria, and alphaproteobacteria are known associated with blooms (Abstract). Earlier published studies include reports where the same taxa are found in different years, including after blooms (admittedly with much less comprehensive overall system analysis than this study).*

We again evaluated the critique of reviewer #2 thoroughly, but continue to disagree with respect to the "quantitative assessment of 'recurring patterns'". We did explain the extent of recurrence and what we deem are patterns. We for sure stand by our interpretation of the data we present. As for novelty, we do not claim that we are the first who found blooms of *Flavobacteriia, Gammaproteobacteria* and *Roseobacter* clade *Alphaproteobacteria* during and after phytoplankton blooms. However, our analysis extends far beyond these high-level taxa and we present high-time resolution data from four consecutive blooms in order to show that many taxa are recurring on the level of genus. To the best of our knowledge, this has not been shown before.

*Specific suggestion: In the figures they should use arrows or other notation to point out the recurring clades and when they bloom.*

All taxa are coded by identical colors throughout years. We tested inclusion of arrows and found the result not convincing. We will work with the *eLife* production team towards the best possible way of visualizing this dataset.